# Linear Mechanisms for Spatiotemporal Reasoning in Vision Language Models

**Raphi Kang**[*], **Hongqiao Chen**[*], **Georgia Gkioxari, Pietro Perona**
California Institute of Technology
{rkang, harrychen, georgia, perona}@caltech.edu

## Abstract

Spatio-temporal reasoning is a remarkable capability of Vision Language Models (VLMs), but the underlying mechanisms of such abilities remain largely opaque. We postulate that visual/geometrical and textual representations of spatial structure must be combined at some point in VLM computations. We search for such confluence, and ask whether the identified representation can causally explain aspects of input-output model behavior through a linear model. We show empirically that VLMs encode object locations by linearly binding *spatial IDs* to textual activations, then perform reasoning via language tokens. Through rigorous causal interventions we demonstrate that these IDs, which are ubiquitous across the model, can systematically mediate model beliefs at intermediate VLM layers. Additionally, we find that spatial IDs serve as a diagnostic tool for identifying limitations in existing VLMs, and as a valuable learning signal. We extend our analysis to video VLMs and identify an analogous linear temporal ID mechanism. By characterizing our proposed spatiotemporal ID mechanism, we elucidate a previously underexplored internal reasoning process in VLMs, toward improved interpretability and the principled design of more aligned and capable models. We release our code for reproducibility: https://github.com/Raphoo/linear-mech-vlms.

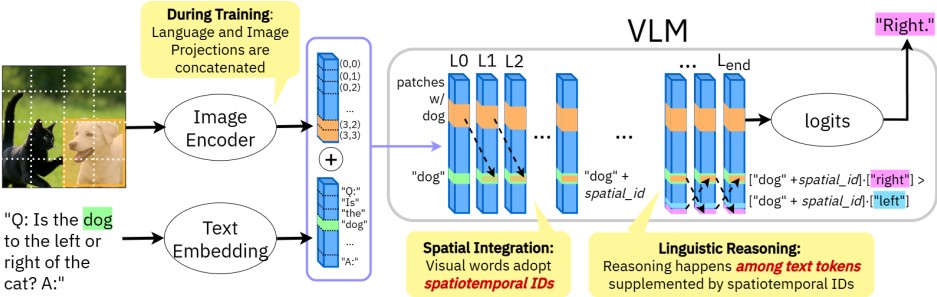

Figure 1: **Hypothesis for spatiotemporal visual reasoning**. The VLM linearly binds spatiotemporal localization to object word activations in early layers. Subsequent linguistic reasoning about the object is informed by its location in space and time per the spatiotemporal ID.

## 1 Introduction

Reasoning about visual input with textual queries is a key challenge behind vision-language models (VLMs). Consider a typical visual question answering (VQA) prompt: "*Is the dog to the left or right of the cat?*". To succeed at this, one must resolve linguistic references, locate entities in the visual field, assess spatial relationships, and make a categorical decision. Though complex capabilities in spatial or temporal reasoning are still far from being fully understood or reliably engineered (Stogiannidis et al., 2025; Chen et al., 2025; Tong et al., 2024), SoTA VLMs have seen steady progress in simple visual reasoning without explicit guidance. So how do they do it?

---

[*]denotes equal contribution

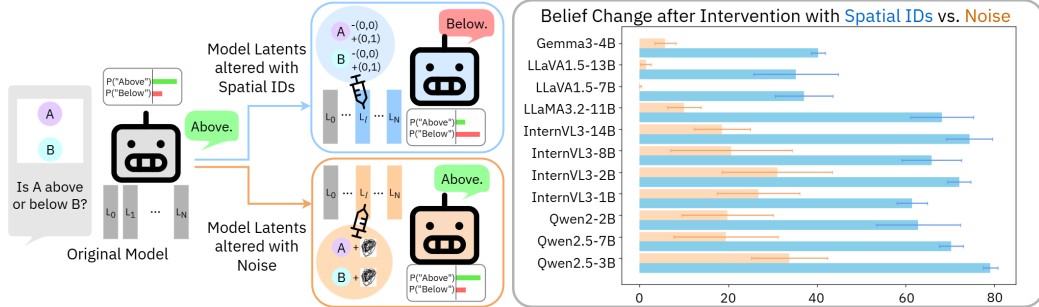

Figure 2: **Results from Targeted Intervention** (§3). Median binary belief swap due to spatial ID steering is 64.4%, and 29.5% for noise. Spatial IDs have 43.6% above-chance influence on average. We conclude that spatial IDs mediate models' beliefs about objects' locations in space.

Attention-based analyses in VLMs have shown various structural properties emerge in VLM internals during VQA (Jiang et al., 2025b; Neo et al., 2024; Zhang et al., 2024a). Relatedly, mechanistic interpretability in LLMs has uncovered linear circuits for relational linguistic reasoning (Park et al., 2024; Feng & Steinhardt, 2024; Hernandez et al., 2024). Might such linear processes also be driving visual reasoning in VLMs, and if so, how exactly? This leads us to ask: **Q1.** *Can we linearly model emergent structured reasoning processes that drive spatial reasoning in VLM internals?*

The typical VLM architecture utilizes a vision encoder which projects the input image to embeddings that are prepended to text token embeddings. This is then processed by a downstream vision-aligned LLM. Popular model families using this paradigm are LLaVA(Liu et al., 2023), LLaMA(Dubey et al., 2024), Qwen (Bai et al., 2025), InternVL (Chen et al., 2024b), and Gemma (Team et al., 2024). A growing body of work aims to improve spatial reasoning capacities in VLMs (Chen et al., 2024a; Fan et al., 2025) and temporal reasoning in video models (Xiao et al., 2024; Li et al., 2024b). Identification of the internal mechanism by which SoTA VLMs do spatial VQA can empower engineers to identify current architectural components leading to VQA failure modes in 3D reasoning or simple VQA. To this end, we ask: **Q2.** *Given our linear model of spatial reasoning in model activations, how do we use it to understand and improve SoTA VLMs?*

Similar training paradigms to image-based VLMs yield video models such as LLaVA-Video(Zhang et al., 2024b), VideoLLaMA3(Zhang et al., 2025), and Qwen2.5 (Bai et al., 2025), among others. Given our theory for the mechanisms underlying spatial reasoning in VLMs, we ask: **Q3.** *Do video models utilize analogous linear mechanisms for temporal reasoning?*

To address these questions, we conduct a mechanistic analysis of autoregressive VLMs and construct a linear model for spatiotemporal reasoning. We show that VLMs decompose a visual reasoning task by first binding spatial information about visual objects to object word activations, in the form of linear components we term *spatial IDs*, answering Q1 (Fig. 1). We then extract these IDs and demonstrate their mediative capacity on model output through targeted belief steering in text activations (Fig. 2). We further find that spatial IDs provide insight on VLMs' struggle with depth reasoning, and incorrect spatial IDs as a result of weak vision encoder or poor modality integration leads to failures in LLaVA and LLaMA. This answers Q2. Finally, we show that temporal IDs similarly mediate video models, answering Q3. In summary, our novel contributions are:

- **Spatial ID Model Formulation**: We propose a linear model of spatial reasoning in VLMs, called *spatial IDs*. These are text-anchored latent structures that bind visual elements to object tokens thus enabling linguistic reasoning about space (§2.1). We emprically extract them from SoTA VLMs for characterization (§2.2).
- **Analytical and Empirical Proof of Causality**: We show model belief can be manipulated by perturbing only the spatial IDs, demonstrating their causal role in reasoning (§3), and provide theoretical intuition for the emergence of spatial IDs in VLMs (§2.3).
- **SoTA VLM Analysis with Spatial IDs**: Through targeted intervention, we identify limitations in depth expression (§4.1) and systematic failure modes in LLaMA/LLaVA (§4.2), and find models can be effectively finetuned with spatial ID guidance (§4.3.).
- **Extension to Temporal IDs in Video Models**: We perform our extraction and characterization analysis on SoTA video models and show that linear temporal IDs, like spatial IDs, can drive temporal reasoning in VLMs (§5).

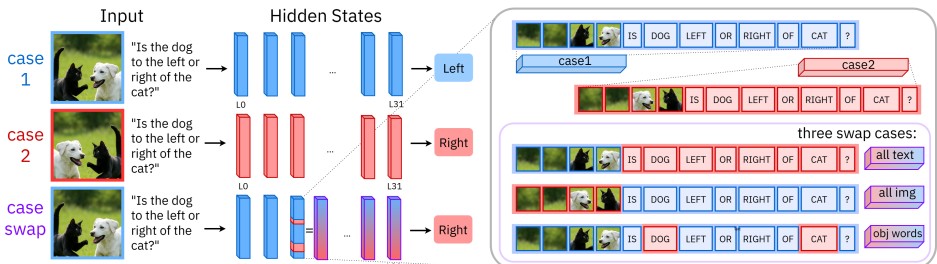

Figure 3: **Mirror swapping experiment** (§2.1). Activations from case 1 and 2 are partially swapped at a select layer, in one of three arrangements. Computations continue normally after this point.

## 2 EMERGENT STRUCTURE IN SPATIAL VISUAL REASONING

In this section, we characterize the spatial reasoning circuits in SoTA VLMs and isolate any linearly separable components used to communicate spatial information. Towards this end, we track information flow in VLMs and identify important junctions for spatial information transfer across token sequences. Then we empirically extract linear spatial IDs, and analytically derive how they arise.

### 2.1 TRACKING INFORMATION FLOW DURING REASONING

To uncover whether VLMs engage in structured visual reasoning, i.e., isolating and propagating spatial information across layers, we intervene on internal activations during inference.

**Mirror Swapping Experiment**. Our goal is to compare the model's output when presented with two distinct images and the same text query. If the model uses localized intermediate representations to reason about spatial relationships, then swapping activations between spatially distinct inputs at key layers and sequence indices should disrupt its final belief about spatial orientation, while swaps between spatially equal but attribute-wise different inputs shouldn't have a strong effect.

Concretely, we run inference on plain and mirrored image-text pairs, extract their representations $x$ at an intermediate layer $L$, then replace a subset $Q$ of activations in the original $x_L$ with activations from the mirrored counterpart $y_L$. The modified representation $\tilde{x}_L$ is passed through the remaining layers. We conduct interventions with three variants of $Q$: (1) all text tokens (2) all image patches (3) object-word tokens only. If information critical to spatial reasoning is concentrated in any of these, the model's belief will change when that region is overwritten. As a control, we concurrently perform "attribute swapping", which follows the same steps but instead of mirroring the input image for the intervention case, changes its colors. The intervention procedure is visualized in Fig. 3 and formally defined in Alg. 1. Further implementation specifics are deferred to Appendix §A.1.

---

**Algorithm 1** Swapping Intermediate Activations in Mirrored Images

| | |
|---|---|
| $x_L, y_L \leftarrow f_L \circ \cdots \circ f_1(x), \quad f_L \circ \cdots \circ f_1(y)$ | ▷ x,y: [seq_dim, embed_dim] |
| $\tilde{x}_L \leftarrow x_L[\tilde{Q}] + y_L[Q]$ | ▷ $\tilde{x}_L$: [seq_dim, embed_dim], Q: [num_of_inds] |
| $\tilde{x}_{\text{out},L}, y_{\text{out}} \leftarrow f_{L_{\max}} \circ \cdots \circ f_{L+1}(\tilde{x}_L), \quad f_{L_{\max}} \circ \cdots \circ f_{L+1}(y_L)$ | ▷ $P_{\tilde{x}_{\text{out},L}}$("GT"): [1] |

---

Here, $Q$ denotes the selected indices in the input sequence to swap, and $\tilde{Q}$ is all other indices. We use the COCO-SPATIAL benchmark (Kamath et al., 2023) for the mirrored images, which is a curated subset of COCO (Lin et al., 2014) annotated with spatial language. To quantify belief shift caused by the intervention, we compute the fraction of the mirror-induced change that can be attributed to the swapped activations at layer $L$. For the ground truth logit "GT", this quantity is derived as:

$$\text{belief shift}_L = \frac{P_{x_{\text{out}}}(\text{"GT"}) - P_{\tilde{x}_{\text{out},L}}(\text{"GT"})}{P_{x_{\text{out}}}(\text{"GT"}) - P_{y_{\text{out}}}(\text{"GT"})} \tag{1}$$

**Results from Mirror Swapping** are shown in Fig. 4A. Through mirror swapping, we observe a *layer-specific effect* for intervention effect across modalities. Intervening on visual patch tokens has a strong effect in early layers but fades with depth. Conversely, interventions on text tokens increasingly affect final outputs in later layers. This is corroborated by observations that middle layers have a modality switching effect in VLMs (Jiang et al., 2025b). Notably, swapping only the object-word tokens alters spatial belief specifically within a narrow band of intermediate layers.

Figure 4: **Ratio change in log probability for logits "left" and "right" from mirror swap** (A) **and attribute swap** (B) **interventions.** (A) shows distinct binary belief swaps, where text tokens have an influence after middle layers. Image patches stop having an influence after that point, and object word tokens *only* have an influence in these middle layers. The control, (B), is noisy.

Attribute swapping results (Fig. 4B) indicate that mirror swapping is a strong experimental setup for assessing spatial information flow in isolation from spurious visual factors. For the belief shift metric, a value of 0.0 on the y axis indicates model belief in the intervened case is equivalent to case 1 (original query), while 1.0 indicates the belief is equivalent to case 2 (mirrored/changed query). Mirror swapping results in distinct and strong binary belief swaps whereas attribute swapping yields mostly noise, to the point belief shift magnitudes are -20∼20x that of the original belief difference.

These results suggest that VLMs extract and encode spatial facts from the image into object word tokens' activations, then operate over them in text-space. We term the latent structures holding visual spatial information *spatial ids*. Inspired by latest mechanistic interpretability findings (discussed in §6), we hypothesize that the manner of spatial information storage is approximately linear.

## 2.2 EMPIRICAL DERIVATION OF SPATIAL IDS

If spatial IDs are indeed linearly bound to object word activations, we should be able to extract them by averaging out object-related semantics from text activations. Below we outline the process of their extraction. In §3, we will test if these IDs causally mediate model beliefs, to validate whether the spatial reasoning mechanism in VLMs is indeed linear.

**Extraction Preliminaries**. We first set up some formalisms to derive spatial IDs. Let $\mathcal{O} = \{o_1, o_2, \ldots, o_N\}$ denote a set of object categories. For each object $o \in \mathcal{O}$, we have a set of images $\{I_{(i,j)}\}$ where the object is positioned at spatial coordinates $(i, j)$ in a $m \times m$ grid. Then let $T^{(o)}$ be a natural language query containing the token corresponding to object $o$, such as "Is there an $o$?". We define $\phi_L(o; I_{(i,j)}^{(o)}, T^{(o)}) \in \mathbb{R}^d$ as the embedding of the text token corresponding to object $o$, extracted from layer $L$ of the VLM when input= $(I_{(i,j)}^{(o)}, T^{(o)})$. The mean embedding for object $o$ at layer $L$ is then:

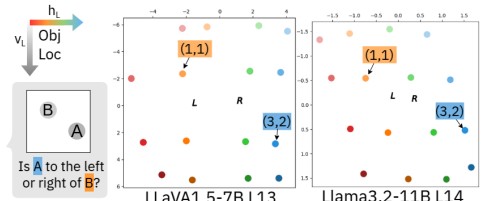

Figure 5: **Spatial IDs in a grid.** Color and saturation of markers represent the location of the object when spatial ID was extracted. x and y axes are coefficients of ID projections onto $h_L$ and $v_L$. L, R represent "left", "right" textual activations.

$$\bar{\phi}_L^{(o)} = \frac{1}{m^2} \sum_{i=0}^{m-1} \sum_{j=0}^{m-1} \phi_L(o; I_{(i,j)}^{(o)}, T^{(o)}) \tag{2}$$

Yielding the object-specific spatial ID at location $(i, j)$ for object $o$:

$$\Delta_L^{(o)}(i, j) = \phi_L(o; I_{(i,j)}^{(o)}, T^{(o)}) - \bar{\phi}_L^{(o)} \tag{3}$$

From this we can derive the *universal spatial ID* at location $(i, j)$, averaged over N objects.

$$\Delta_L(i, j) = \frac{1}{N} \sum_{n=1}^{N} \Delta_L^{(o_n)}(i, j) \tag{4}$$

To extract canonical horizontal and vertical directions from the universal spatial IDs $\Delta_L(i, j) \in \mathbb{R}^d$, we compute average difference vectors across grid-aligned coordinate pairs. The vertical and horizontal direction vectors $v_L, h_L \in \mathbb{R}^d$, corresponding to increasing $i$ and $j$, are computed based on the spatial IDs. Eq. 5 shows the derivation for $v_L$, and $h_L$ is derived in an analogous manner.

$$v_L = \frac{1}{m \cdot \binom{m}{2}} \sum_{i=0}^{m-1} \sum_{j_1 > j_2} [\Delta_L(i, j_1) - \Delta_L(i, j_2)] \tag{5}$$

**Empirical Extraction**. For our study, we extract spatial IDs from 11 SoTA VLMs, with synthetic images created from open-source OBJAVERSE (Deitke et al., 2023) objects. The object renders are paired onto a grid of $m = 4$ on top of random natural backgrounds. We provide further extraction details in Appendix §A.2, along with ablations showing extracted spatial IDs are invariant to chosen images §D and counterfactual studies confirming that spatial IDs reside in object words, and spatial axes are orthogonal §C. Fig. 5 shows two example spatial ID grids projected onto their respective spatial vectors. IDs from more models are shown in §B. We see that these extracted IDs arrange in an approximate $m \times m$ grid at modality binding layers. Also projected are activations for spatial words, where we find that "left" is closer to leftmost spatial IDs, and "right" vice versa.

### 2.3 THEORETICAL SKETCH OF SPATIAL IDS

We now offer a quick, highly minimal analytical intuition for how the emergence of spatial IDs can be ubiquitous across many different models. Let $p = (i, j)$ be some coordinate on a $m \times m$ grid. Then for some query to a VLM, let the input sequence contain projected visual tokens $\{x_p\}$ for all $p$, and the query text tokens include an object token $o$. The residual update to $o$ by one head is:

$$r_o \leftarrow r_o + W_{\text{out}} \sum_{p \in P} \alpha_{o \leftarrow p} \, v_p, \qquad \alpha_{o \leftarrow p} \propto \exp\left(\frac{q_o^\top k_p}{\sqrt{d}}\right), \quad v_p = W_V x_p. \tag{6}$$

With cross-modal alignment, attention peaks at the true object patch $p^\star$, giving $\delta r_o \approx W_{\text{out}} W_V x_{p^\star}$. Decompose each patch as $x_p = s_p + P \psi(p) + \varepsilon_p$, where $s_p$ encodes content, $\psi(p) \in \mathbb{R}^{d_\psi}$ is a shared positional basis (e.g. RoPE or learned 2D embeddings), $P$ maps positional features into model space, and $\varepsilon_p$ is small. We can now substitute $\phi_L(o; I_{p^\star}, T^{(o)}) = r_o + \delta r_{o,p^\star}$ into Eq. 3. A detailed derivation is in §2.2, but in summary we get:

$$\Delta_L(p^\star) = \Delta_L(i, j) \approx \underbrace{W_{\text{out}} W_V P}_{M \text{ (fixed per model)}} \left(\psi(i, j) - \tfrac{1}{m^2} \sum_p \psi(p)\right). \tag{7}$$

Thus, spatial IDs are approximately a linear transformation of a universal positional basis written into the object token by attention. Spatial logits are thus approximately linear readouts:

$$\ell(\text{LEFT}) - \ell(\text{RIGHT}) \approx (w_{\text{LEFT}} - w_{\text{RIGHT}})^\top \Delta_L(i, j), \tag{8}$$

so if $(w_{\text{LEFT}} - w_{\text{RIGHT}})^\top M$ aligns with the $x$-coordinate in $\psi$, the model prefers "left." Empirically, a low-rank linear fit from positional encodings $\psi$ to spatial IDs $\Delta_L$ explains most variance (e.g. rank-3 gives $R^2 \gtrsim 0.85$, see §E.2, Table 1). A more detailed derivation for $\Delta_L(i, j)$ for the multihead case is shown in Appendix §E.1. This is a particularly simplified setting, and real reasoning circuits in VLMs will involve a lot more noise and nonlinearities. The main takeaway is that VLM designs like Fig. 1 encourage models to endow text tokens with visual information, followed by linguistic reasoning. This information transfer, in its most simplified linear form, is in the form of spatial IDs.

In practice, the finegrained circuit employed by VLMs may be much more varied, distributed, and nonlinear. The spatial ID framework could capture just one component of a more complex system. But per Ockham's Razor, spatial IDs are powerful due to their simplicity. In following sections, we demonstrate the mediative influence of this simple spatial ID model on final VLM outputs, and further show how spatial IDs can be leveraged to improve existing models and build stronger ones.

## 3 SPATIAL IDS MEDIATE MODEL BELIEFS

If spatial IDs capture the causal mechanisms behind spatial reasoning, we should be able to linearly subtract or add arbitrary IDs to object word activations and change the model's belief about object location. In this section, we design and perform experiments on real naturalistic images to test that empirically derived spatiotemporal IDs have causal effects on model outputs on spatial VQA.

**Steering with Arbitrary IDs Experiment**. For some layer $L$, we denote the model residuals corresponding to the entire input sequence after that layer as $x_L$, and perturb its token activation at some index $q$ to observe any effects on the output belief. Alg. 2 illustrates the process.

Here we scale the norm of $\Delta_L(i, j)$ to be $\alpha |x_L[q]|$, and $\tilde{\Delta}_L(i, j) = \Delta_L(m - i - 1, j)$. This approximately preserves the norm of $x_L$. $\alpha = 5$ is some scaling constant set after grid searching for stable intervention. We take 100 COCO images where one object is to the left or right of another, per labels from COCO-SPATIAL, and ask queries of the form "Is x to the left/right of y"?. We measure the log probability of "left" and "right" tokens in the final output logits to assess steering effects.

---

**Algorithm 2** Intervention at Layer $L$ via Residual Modification

$x_L \leftarrow f_L \circ \cdots \circ f_1(x)$       $\triangleright$ x: [ seq_dim, embed_dim]

$\tilde{x}_L \leftarrow x_L[: q] + [x_L[q] + \Delta_L(i,j) - \tilde{\Delta}_L(i,j)] + x_L[q+1 :]$       $\triangleright \Delta_L(i,j)$: [embed_dim]

$\tilde{x}_{\text{out}} \leftarrow f_{L_{\max}} \circ \cdots \circ f_{L+1}(\tilde{x}_L)$       $\triangleright P_{\tilde{x}_{out}}(\text{"GT"})$: [1]

---

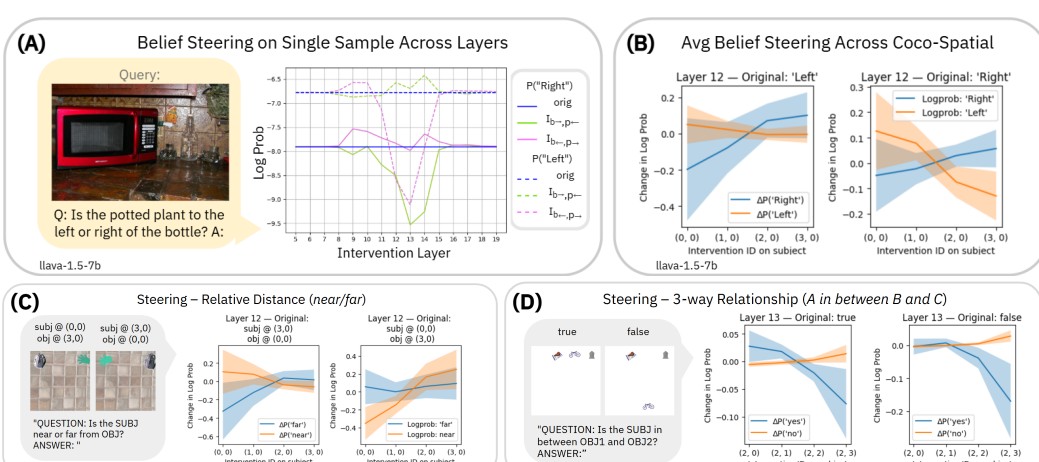

Figure 6: **Effect of spatial steering on real images** on one sample across different intervention layers (A) and across a dataset for one layer (B). In (A), dotted and solid lines indicate answer probabilities for "left" or "right". Different colors indicate no intervention (blue), steering the bottle to the left and plant to the right (pink), and the reverse steering (green). Blue lines are flat and show that the unintervened model incorrectly assigns a higher log probability to "left". Pink lines show intervention on intermediate layers results in overwriting initial incorrect beliefs. (B) shows the shift in log probability for "'left" vs. "right" as a result of spatial steering on the subject word token. (C) and (D) show shifts in log probability for "near" vs. "far", and "yes" to an object being sandwiched between two others, vs. "no".

**Results from Arbitrary Steering**. Fig. 6 shows the effects of model belief steering on real images and videos. Fig. 6A shows that steering impact is greatest at modality alignment layers as expected per the mirror swapping analysis, and Fig 6B shows that intervening with the rightmost spatial ID largely enhances model belief that the object is to the right, and vice versa for the leftmost ID for leftward belief. The y axes show changes in log probability for those binary directions for the whole dataset, and x axes show the different ID locations. Regardless of whether the answer to the original query was "left" or "right", subplot trends are the same.

We repeat the analysis for queries about relative distance and three-way relationships where one object is sandwiched *in between* two others. Again, we find that when the object is to the left, altering the spatial ID of the subject towards the right increases the likelihood of "far" and decreases that of "near", and vice versa if the object is to the right. Similarly, we find that bringing a subject closer and closer to be surrounded by two objects increases the model's belief that the subject is *in between* the objects.

**Adversarial Steering Experiment**. If spatial IDs are indeed ubiquitous across models, interventions on internal activations should change the resultant model beliefs across many SoTA models. To confirm this, we evaluate the log probability of the correct answer ("GT") and its opposite ("¬GT") for all samples in COCO-SPATIAL on 11 SoTA models. Then, we repeat this measurement after intervention with spatial IDs most likely to reverse their original beliefs. More detailed experimental procedure is provided in §A.5. In addition to targeted adversarial steering, we perform steering with noise vectors of the same norm as the spatial IDs, to evaluate chance belief swaps.

**Adversarial Steering Results**. We report % binary belief swaps on COCO-SPATIAL from the spatial ID vs. noise steering case in Fig. 2. Steering with spatial IDs yields a median 64.6% swap in beliefs, versus 29.5% with noise. This indicates activation intervention has nonzero chance influence on model output, but there is a clear above-chance average of 43.6% increase with spatial IDs. Here, a model's belief on one sample is considered "swapped" if the relative likelihood of the ground truth

and its opposite answer has changed. For example, if $P(\text{"left"}) > P(\text{"right"})$ before intervention, but after intervention we see $P(\text{"left"}) < P(\text{"right"})$, the intervention has swapped the model belief. Thus we conclude that spatial ID mechanisms mediate model belief in the models considered.

## 4  Spatial IDs for Understanding and Improving Image VLMs

With the existence and causal nature of spatial IDs established, we explore two ways to leverage them towards stronger VLMs. First, we aim to understand why 3D reasoning fails in SoTA VLMs. Second, we use spatial IDs to diagnose architectural bottlenecks of SoTA VLMs in VQA.

### 4.1  Depth Representation in Image VLMs

Spatial IDs suggest that VLMs represent visual space within a 2D grid. What might this mean for depth? We hypothesize that the language model must reason about depth related queries using the 2D localization in context. To verify whether this is the case, we look at the resulting belief changes in the depth axis when the LLaVA1.5 7B model is steered with spatial IDs varying in height. Fig. 7 shows the results. The same spatial IDs increasing the likelihood for "above" and decreasing "below", also drive up "front" and drive down "behind" in LLaVA.

Further, projection of these word embeddings onto spatial vectors reveals that "above"/"behind" and "below"/"front" map to overlapping locations, indicating their functional relationships with spatial IDs are similar. These results may be due to biases in training, or innate shortcomings in the VLM architecture. They certainly highlight the need for better depth-handling mechanisms, whether that be through improved training data or tooling.

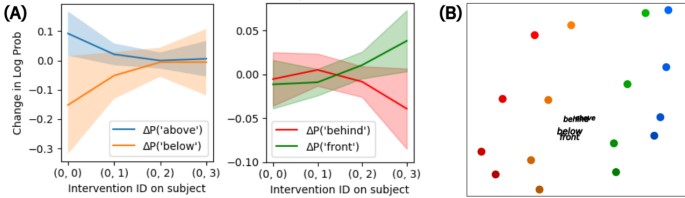

Figure 7: **Depth and height are strongly correlated in LLaVA**. (A) Steering results for IDs varying in y-dim and their impact on beliefs about height or depth. (B) Projection of spatial words onto a spatial ID grid. Embeddings for "above"/ "front" and "behind"/"below" are nearly identical.

### 4.2  Diagnosing VLMs

When a VLM fails at a spatial task, how do we pinpoint the reason it failed? Referring back to Fig. 1, VLM failure points can roughly be divided into modality encoding, crossmodal information integration, or linguistic reasoning stages. Knowing what part of a VLM's architecture must be improved to reduce failures is paramount to efficient model engineering.

Per-sample analysis of spatial IDs provides a unique ability to identify a model's bottleneck. Consider an evaluation set $\mathbf{K} = \{k_1, k_2...k_K\}$, where each $k = (image, query)$. An imperfect VLM will fail at some samples. In this section, we perform two experiments to identify the architectural component which causes for the distribution of $\mathbf{K}_{wrong}$ to be statistically distinct from $\mathbf{K}_{correct}$.

An example diagnosis process may look like this. If a model exhibits *incorrect* spatial ID binding, and that incorrect output produced is faithful with the spatial ID, then the language-only reasoning stage is likely not at fault. From there, if a model exhibits sensitivity to masking the correct object region for $\mathbf{K}_{wrong}$ but not for $\mathbf{K}_{correct}$, the vision encoder is the likely bottleneck. If there is no distinct sensitivity difference, the errors are likely taking place after the vision encoder, but before the linguistic reasoning. If model accuracy seems independent of both spatial ID correctness and image recognition capacity, the language model layers beyond spatial ID binding are likely the biggest bottleneck. Note that it is possible for incorrect spatial IDs to be correlated to wrong answers, but still have some model inaccuracies be resultant from factors other than spatial IDs, such as erroneous priors during LM readout (Leng et al., 2024; Ramakrishnan et al., 2018). In this case, it is still valuable to find if models can benefit from stronger spatial representations through this diagnosis process, and minimize avenues for failure. For the described analyses, we need a sufficient $\mathbf{K}_{wrong}$ subset. As their $\mathbf{K}_{wrong}$ are biggest on COCO-SPATIAL, we select LLaVA1.5 7B and LLaMA3.2VL 11B as model organisms for this section.

**Ground Truth Spatial ID Deviation Experiment**. First, we want to identify if models predict incorrect spatial IDs for the samples they get wrong. If the answer is *yes*, then it is likely that the

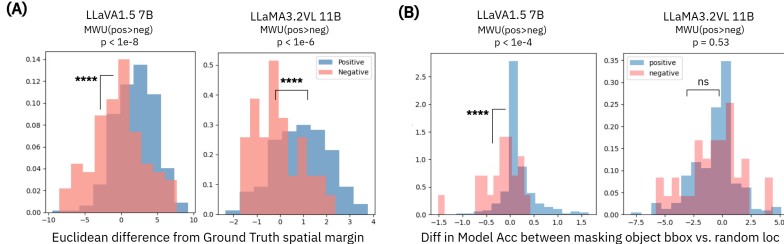

Figure 8: **Contrasting density histograms show incorrect spatial IDs drive bad predictions**. (A) shows deviation of model spatial IDs from g.t., and (B) the difference in model accuracy when masking objects vs. random locations in images. Histograms are samples VLMs got right (blue) or wrong (red). LLaVA shows faulty object detection with wrong answers, while LLaMA does not.

downstream language model is not the performance bottleneck, since it is faithful to the spatial information received. To compute the deviation of the model's believed spatial ID to the ground truth (g.t.), we compute the g.t. spatial ID by projecting the word activation onto the spatial axes:

$$\Delta_L^{(o)}(i,j)_{ext} \approx VV^T\phi_L(o; I_{(i,j)}^{(o)}, T^{(o)}), \quad V = [v_L, h_L] \tag{9}$$

For a spatial query like "Is the $o$ to the left or right of a $\tilde{o}$?", we can thus compute $\Delta^{(o)}(i,j)_{\text{gt}}$ and $\Delta^{(\tilde{o})}(i,j)_{\text{gt}}$. The model's assigned spatial IDs to the objects are computed per Eq. 9, for $\Delta^{(o)}(i,j)_{\text{ext}}$ and $\Delta^{(\tilde{o})}(i,j)_{\text{ext}}$. Then the g.t. ID margin deviation for some object $o$ is:

$$\text{ID deviation margin} = \epsilon_{\text{ext}} - \epsilon_{\text{gt}}, \quad \text{where } \epsilon_{\text{gt}} = i_{\text{gt}}^{(o)} - i_{\text{gt}}^{(\tilde{o})}, \epsilon_{\text{ext}} = i_{\text{ext}}^{(o)} - i_{\text{ext}}^{(\tilde{o})} \tag{10}$$

Here, a negative margin indicates that the model's extracted spatial IDs oppose the ground truth.

**ID Deviation Results**. From Fig. 8A, we see that deviation from ground truth in extracted spatial ID margin is highly correlated with model mistakes. In other words, for LLaVA and LLaMA, wrong spatial IDs in object word activations led to wrong model answers, so linguistic reasoning was not the reason these failures occurred. Each subplot shows two density histograms overlaid in the same grid, where the x axis is $\epsilon_{\text{ext}} - \epsilon_{\text{gt}}$. The red histogram represents the density of ID deviations for $\mathbf{K}_{wrong}$, and the blue histogram shows the same for $\mathbf{K}_{correct}$. The red distribution is visibly skewed to the negatives compared to the blue. Quantitatively, we perform the Mann-Whitney U test (McKnight & Najab, 2010) to calculate the p-value for the hypothesis that the two distributions (red and blue) are non-identical. Now we ask, is this failure mode stemming from the vision encoder level, or does it occur during the spatial ID binding across modalities?

**Image Masking Experiment**. Altering the raw image input at the pixel level can help us understand whether it is a faulty vision encoder or faulty crossmodal information integration that has led to the failures. If the model's beliefs on $\mathbf{K}_{correct}$ are more sensitive to masking the image raw input at the g.t. location of $o$, while beliefs on $\mathbf{K}_{wrong}$ change more with masking elsewhere, we can conclude that the vision encoder of this VLM is doing a poor job at object detection, leading to observed failures. If we do not observe this is the case, the failure may arise from the crossmodal information integration stage. In other words, the language model is doing a poor job appending binding IDs, despite the vision encoder having the necessary object recognition capacity.

We design an obfuscation paradigm inspired by methods like D-RISE (Petsiuk et al., 2021), where we either blur the bounding box of $o$, or $R$ other locations in the image that do not intersect with the bboxes for $o$ or $\tilde{o}$. We then measure model belief change when masking the object vs. elsewhere:

$$\text{bbox sensitivity} = \big(P(\text{"GT"}) - P(\text{"GT"}|\text{mask } o)\big) - \big(P(\text{"GT"}) - min_r[P(\text{"GT"}|\text{mask } r), r \in R]\big) \tag{11}$$

**Image Masking Results**. Fig. 8B shows overlaid histograms for bounding box masking sensitivities of $\mathbf{K}_{correct}$ and $\mathbf{K}_{wrong}$. Here, a negative value indicates greater sensitivity to raw pixel masking of random scenes, suggesting poor object detection. For LLaVA, there is a statistically significant p-value for the hypothesis that $\mathbf{K}_{wrong}$ is shifted more negative than $\mathbf{K}_{correct}$, indicating its vision encoder fails at object detection when it answers incorrectly. In contrast, $\mathbf{K}_{wrong}, \mathbf{K}_{correct}$ in LLaMA are agnostic to image obfuscation. This suggests that its failure modes likely stem after the vision encoder. These insights could be attributed to how LLaVA uses an out-of-the-box ViT that

was text-aligned at a massive scale, hence not being tuned for finegrained detection, while LLaMA has a trained in-house ViT whose image-text alignment may be less robust.

**Diagnosis Conclusion**. With spatial IDs, we explore the causes for failure in a few model VLMs. We find that for both LLaMA and LLaVA, the linguistic reasoning stage is faithful to spatial IDs. LLaVA's vision encoder is likely creating wrong spatial IDs from poor object detection, while LLaMA's weak point appears to be information integration across the image patch activations to the text tokens. These conclusions are preliminary and do not suggest that *all* of a model's failures stem from *one* architectural component, but can serve to guide finetuning stage choices when resources are scarce, or provide intuition for future model designs.

## 4.3 Improving VLMs

**Spatial IDs and Model Performance**. To understand if spatial IDs could be a valuable learning signal, we first evaluate whether stronger steerability from spatial IDs is correlated to stronger models. Fig. 9 shows the results of this analysis, where indeed we see that models with higher zero-shot accuracy on COCO-spatial also exhibit greater belief changes with spatial ID interventions.

We define "steerability" as the difference between the change of belief resultant from steering with opposing spatial IDs versus noise. The layers of intervention are chosen as the middle third of all layers for that model. Each point shows the model's mean steerability (on x) against its accuracy on COCO-spatial with no spatial intervention. Dotted lines connect models within a family.

**Spatial Loss Module.** Fig. 9 shows spatial IDs signal stronger model performance. This suggests that the strength of spatial IDs could be a valuable learning signal for VLMs to learn principled spatial reasoning. To validate this intuition, we finetune Qwen2-2B on a synthetic dataset similar to the one used to extract spatial IDs, and evaluate on COCO-Spatial. We introduce an additional

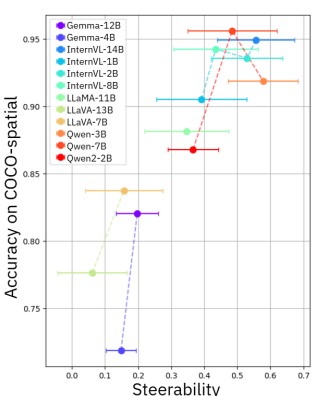

Figure 9: **Accuracy vs. Steerability**. Models with higher accuracy can be better steered with spatial IDs.

loss module at layer 11 that computes the cosine similarity between the predicted and ground-truth spatial ID at that layer. We provide detailed explanations for this process in §A.9. This spatial ID loss is added to the standard language modeling objective, providing extra supervision. We perform a control training without the spatial ID loss. As the training, and thus the spatial alignment, was performed on simplistic synthetic data, both models saturate and start to overfit after reaching peak validation accuracy around 90%. But we see that explicit spatial ID loss helps the model generalize faster, reaching 91% accuracy on COCO-spatial at 3.2k steps, a net 6% accuracy gain over the control case under the same number of training steps.

|  | Num Steps | 0 | 800 | 1600 | 2400 | 3200 |
|---|---|---|---|---|---|---|
| **Control** | LM Loss ($\downarrow$) | 3.30 | 0.05 | <0.01 | <0.01 | <0.01 |
|  | COCO Val Accuracy ($\uparrow$) | 0.77 | 0.83 | 0.84 | 0.85 | **0.85** |
| **With Spatial Loss** | LM Loss ($\downarrow$) | 2.71 | 0.04 | <0.01 | <0.01 | <0.01 |
|  | Spatial ID Loss ($\downarrow$) | 0.75 | 0.58 | 0.41 | 0.36 | 0.33 |
|  | COCO Val Accuracy ($\uparrow$) | 0.77 | 0.83 | 0.84 | 0.88 | **0.91** |

## 5 Temporal IDs in Video Models

Thus far, we have characterized spatial IDs as a causal model for spatial visual reasoning in VLMs. Could we find a similar linear paradigm for the temporal axis? In this section, we repeat the experiments in §2-3 for the temporal dimension in video models, with the goal of identifying linearly separable temporal markers on object words. For space, experimental procedures are described briefly here, and in greater detail in Appendix §A.

### 5.1 Mirroring, Extracting, and Steering across the Temporal Axis

**Temporal Mirror Swapping**. We validate that there exist modality alignment layers with object-level visual information transfer in video models. For mirrored videos, we take the Scene_QA subset

of MVBENCH (Li et al., 2024a) and swap the order of frames from back to front. Following Alg. 1, we show results for swapping text tokens, image patches, and object words in Fig. 10A. While the error bound is noisier than spatial LLaVA, likely as LLaVA-Video follows response formats less well, we see the expected bump around middle layers for crossmodal integration.

**Temporal ID Extraction**. Derivation of temporal IDs and the temporal vector $t_L$ follows Eq. 2 - 5, with synthetic 8-frame videos of OBJAVERSE renders. Results are shown in Fig. 10B. We again see that the text activation for "before" projects closer to earlier frames, than the activation for "after".

**Causality of Temporal IDs**. Finally, to confirm controllability with arbitrary temporal IDs, we perform the steering experiment per Alg. 2 on MVBENCH videos. Results are shown in Fig. 10C. On these real, naturalistic videos, we see that later temporal IDs steer the model belief towards "after", and earlier IDs towards "before", as expected.

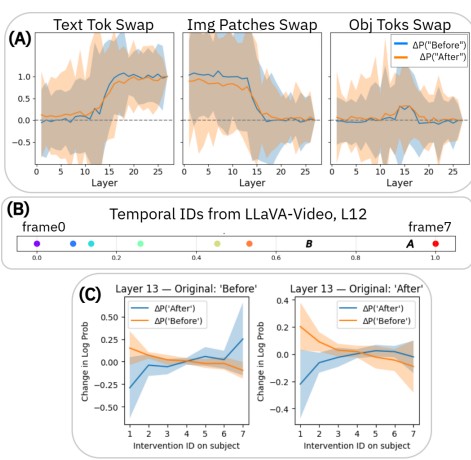

Figure 10: **Temporal ID Results**. Mirror Swapping on videos (A), Temporal ID grid (B), and temporal ID steering on model beliefs (C) with LLaVA-Video

### 5.2 EMERGENCE OF TEMPORAL IDs

Fig. 10 shows summary results on LLaVA-Video, but we include temporal IDs from VideoLLaMA3 and Qwen2.5 in Appendix §B.2. LLaVA-Video and VideoLLaMA3 use textual description of the video length and number of frames to indicate timestamps preceding the visual input, while Qwen uses explicit MRoPE time IDs. This suggests that spatiotemporal IDs can emerge without explicit positional encoding, beyond the simple mechanism derived in Eq. 7.

## 6 RELATED WORK

Mechanistic interpretability is a growing field uncovering the inner workings of large models, popularizing techniques such as circuit tracing (Elhage et al., 2021; Ameisen et al., 2025), Sparse Autoencoders (Cunningham et al., 2023), linear probing (Alain & Bengio, 2016), and others. The Linear Representation Hypothesis posits that concepts are linearly encoded in LLM latents (Park et al., 2024), and activation patching supports that linear changes in activations drive model belief (Meng et al., 2022; Zhang & Nanda, 2023). Internal in-context reasoning mechanisms such as linear *binding IDs* (Feng & Steinhardt, 2024; Feng et al., 2024) have been identified in LLMs, along with other evidence for linear multi-hop reasoning (Yang et al., 2024), in-context task vectors (Hendel et al., 2023) and linear relational embeddings (Hernandez et al., 2024) during reasoning.

Linearity of embeddings have also been discovered in VLM latent spaces (Trager et al., 2023; Jiang et al., 2025a) to some degree. Previous work showed that VLMs separate VQA into image-focused then text-focused stages (Jiang et al., 2025b), and others have extended LLM interpretability techniques like logit lens (Neo et al., 2024) or attention tracking (Zhang et al., 2024a; Yu & Ananiadou, 2025) to VLMs to unearth internal circuits. In our work, we mechanistically capture spatiotemporal information flow from image patches to text tokens in VLMs, via the spatial ID mechanism.

## 7 CONCLUSION, LIMITATIONS, & FUTURE WORK

We propose spatiotemporal IDs as a linear model for visual reasoning about space and time in VLMs. With a series of causal analyses, we show these IDs can be obtained in many SoTA models, and that they closely mediate models' beliefs about visual objects' location in space and time. We further offer ways to extend this mechanistic insight to improving existing VLMs. For tractability, our work is currently limited to analyses in simple spatial queries or appearance-based temporal queries. Experimental design for more complex, open-ended queries will enhance our understanding of how VLMs utilize rudimentary concepts like spatial IDs in more nuanced settings. Further, we only extract and steer models of sizes up to 14B parameters. Investigation into whether the spatial ID circuit plays a similarly prominent role in larger models will reveal whether VLMs of varying capacities follow analogous methods for visual reasoning, or employ distinct measures. Lastly, while we show several potential ways to leverage spatial IDs for VLM diagnostics or finetuning, future work could include expanded use cases such as explicit temporal guidance at large scale.

## ACKNOWLEDGMENTS

We would like to thank Aadarsh Sahoo, Jiahai Feng, Michael Hobley, and Brian Cheung for valuable discussions and proofreading. Raphi Kang is funded by the NSF Graduate Research Fellowship. Georgia Gkioxari is supported by the Hurt Scholar program, Meta and Google. Hongqiao Chen was supported by the Caltech SURF program.

## REPRODUCIBILITY

We provide finegrained details for all experiments in §A of the Appendix, and results on all the models considered in §B. We include experimental details, results from various ablation analyses and counterfactual trials in §C-D. Code for all experiments can be found at: https://github.com/Raphoo/linear-mech-vlms.

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
