LINEAR MECHANISMS FOR SPATIOTEMPORAL REASONING IN VISION
LANGUAGE MODELS
–SUPPLEMENTARY MATERIAL–

# A EXPERIMENTAL DETAILS

## A.1 MIRROR SWAPPING

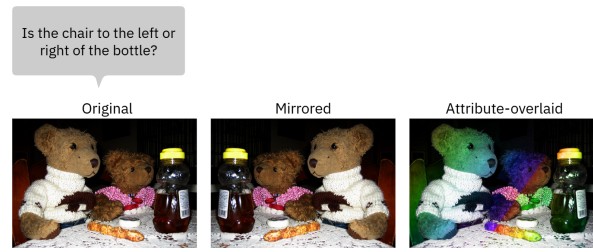

Figure A1: Example altered images for mirror swapping and attribute swapping.

**Token handling**. Different models and tokenizers have different tokenizing schemes. For example, for the query "Question: Is the the thermometer to the left or right of the desktop? Answer left or right. Answer: ", the tokenization from Gemma, LLaMA, LLaVA, and Qwen will be as shown:

```
=== Gemma ===
Tokens: ['Question', ':', '_Is', '_the', '_thermometer', '_to',
'_the', '_left', '_or', '_right', '_of', '_the', '_desktop', '?',
'_Answer', '_left', '_or', '_right', '.', '_Answer', ':']

=== LLaMA ===
Tokens: ['_Question', ':', '_Is', '_the', '_therm', 'ometer',
'_to', '_the', '_left', '_or', '_right', '_of', '_the', '_desktop',
'?', '_Answer', '_left', '_or', '_right', '.', '_Answer', ':']

=== LLaVA ===
Tokens: ['_Question', ':', '_Is', '_the', '_therm', 'ometer',
'_to', '_the', '_left', '_or', '_right', '_of', '_the', '_desktop',
'?', '_Answer', '_left', '_or', '_right', '.', '_Answer', ':']

=== Qwen ===
Tokens: ['Question', ':', 'ĠIs', 'Ġthe', 'Ġthermometer', 'Ġto',
'Ġthe', 'Ġleft', 'Ġor', 'Ġright', 'Ġof', 'Ġthe', 'Ġdesktop', '?',
'ĠAnswer', 'Ġleft', 'Ġor', 'Ġright', '.', 'ĠAnswer', ':']
```

When a word is represented as multiple tokens per a model's processor (e.g., $frog$ is tokenized into $[\_f, rog]$ in LLaVA), we take the last index of this list to be most representative of the object, as it is the distinguishing element. So in the case of LLaMA or LLaVA, we would take the "ometer" token to represent the object "thermometer".

**Logit Probabilities**. For assessing the model's likelihood of saying "left" vs. "right", or two other options, we take the log probability for that token following the tokenization scheme of the model family. This means we take the model output.logits and index at the token ID for 'Ġright' in Qwen, for example, to get P("right").

**Activation Patching**. For every model, we first register a forward hook at each layer to collect the intermediate activation for both the original (case 1) and mirror-swapped (case 2) cases. Then, we register another forward-hook replace the original activations with the mirror-swapped one at select indices according to the three different settings.

## A.2 SPATIOTEMPORAL ID EXTRACTION

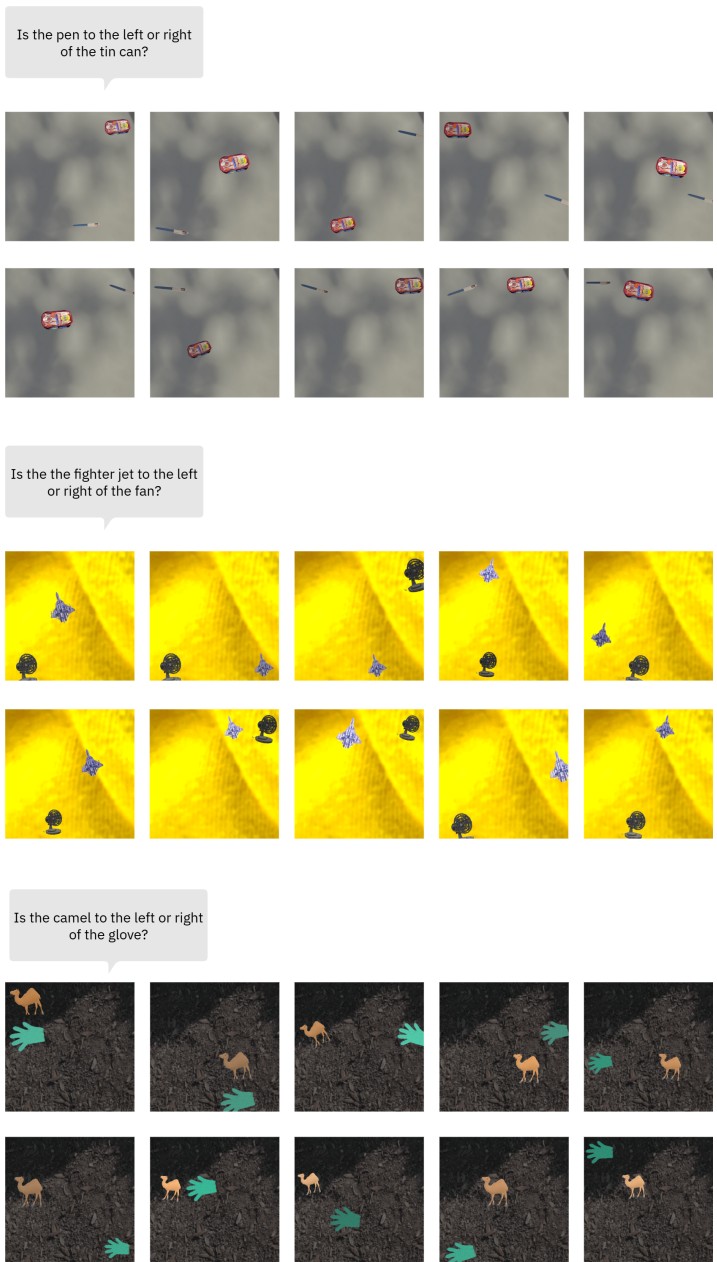

Figure A2: Synthetic images used towards spatial ID extraction

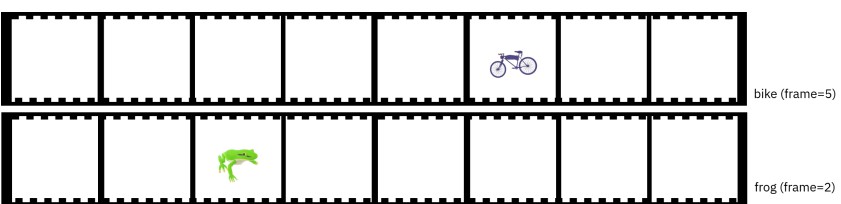

Figure A3: Illustration of synthetic videos used towards temporal ID generation. All videos had 8 frames.

**Synthetic Image generation for Spatial IDs**. We use 55 OBJAVERSE object renders and project them in various pairs onto random backgrounds, per (Kang et al., 2025). All images with the same objects get the same text query. In §D.1 we show the different number of objects and total number of images used to generate spatial IDs. For $w$ object pairs, we generate $w \times s \times m^2 \times (m^2 - 1)$ total images, where $s$ is the number of object sizes we consider, and $m$ is grid size. While we find minimal difference with extraction dataset size, as shown in §D.1, we use 90 object pairs, and consider $s = 4$ from $\{224, 174, 124, 74\}$, yielding 86,400 images. Note that each image size is $224 \times 4 = 896$ in width and height.

**Synthetic Video generation for Temporal IDs**. We take 5 unique OBJAVERSE object pairs in 61 distinct temporal arrangements. For baseline temporal ID extraction, all objects were centered in the image. For spatial vs. temporal disentanglement verification, we try three spatial variants - left, center, and right - for object location.

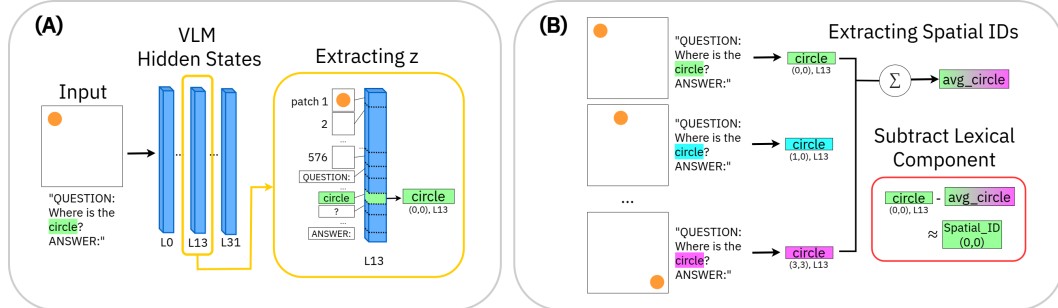

Figure A4: Illustration of spatial ID extraction. We isolate the relevant visual object word token in a chosen layer activation (A) and compute the shared lexical component for that particular object word that is independent of spatial localization (B) to acquire the linearly bound spatial ID.

### A.3 ARBITRARY STEERING EXPERIMENTS

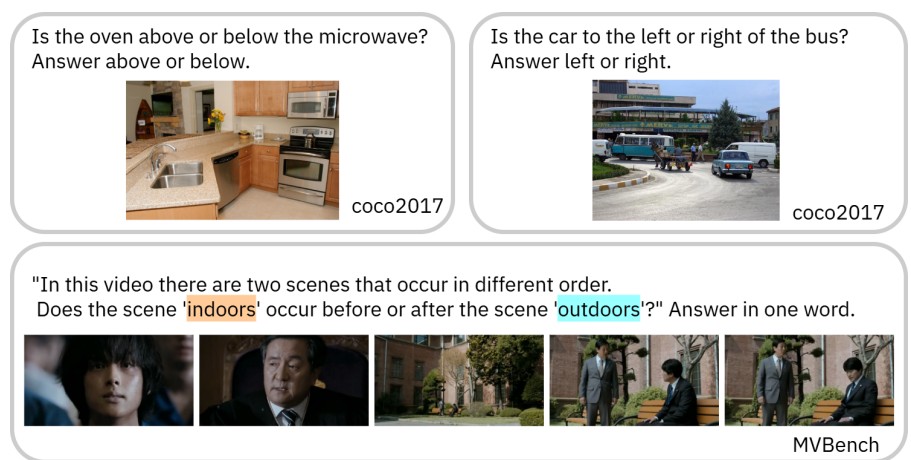

Figure A5: Examples of real images (top) and videos (bottom) we use to test model beliefs.

### A.4 COLOR-BINDING REASONING EXPERIMENTS

Do spatial IDs mediate visual reasoning beyond direct spatial queries (such as A above/below B, etc.)? To test this, we perform mirror swapping on two images where two objects are *in the same place*, unlike the mirror swapping in §2.1. This time, the objects are opposing in color. Fig. A6 shows the example query setup, as well as the results of swapping all image tokens, all text tokens, just the color word tokens, or just the object word token.

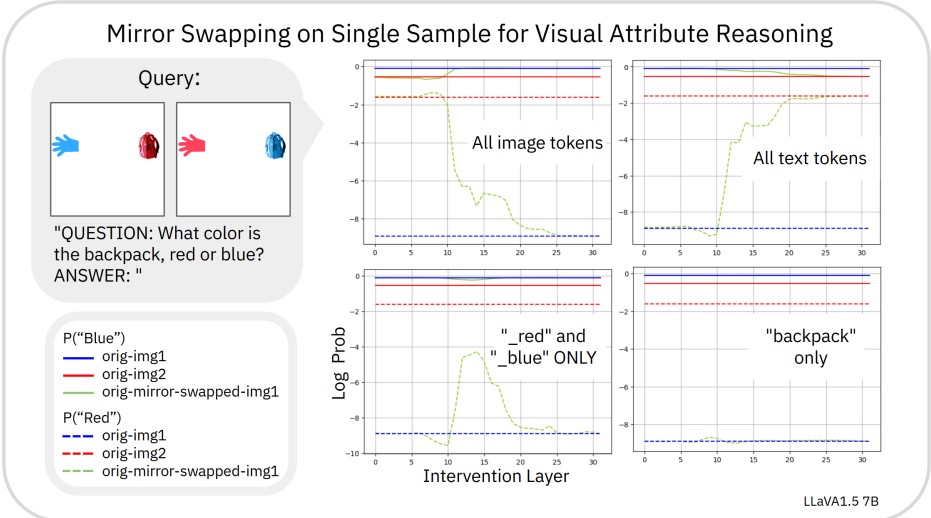

Figure A6: Mirror Swapping on Single Sample for Color Binding

Notice that swapping the activations for "backpack" has no effect, since the spatial ID encoded in the object word activation stays the same regardless of the input image (the backpack is in the same location in both images). Swapping the activations of color-related words, on the other hand, alters model belief at key modality binding layers. This suggests that the color words were storing spatial IDs that corresponded to the location where that color was present, and matching this color spatial ID to the object token was the readout process.

We repeat this experiment across 100 total such images, and show the results in Fig. A7. On average we see that swapping the color word tokens influences model beliefs in intermediate layers, much more so than swapping non-color word tokens. This suggests that spatial IDs mediate visual reasoning beyond direct spatial queries.

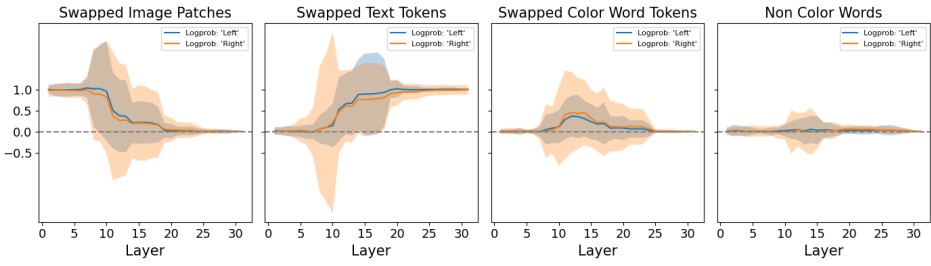

Figure A7: Swapping tokens for color binding.

## A.5 ADVERSARIAL STEERING EXPERIMENTS

We perform steering on layers 9 through layer 2(model_len // 3) per model. To ensure activation norms don't explode, we test a few different scaling factors for the interevning spatial ID's norm. In the scaling factor = 1 case, we scale the norm of the spatial ID to equal the norm of that word token's activation vector. We try a few scaling factors and choose 5 for steering all models, both for the noise vectors as well as the spatial IDs. Here, the norm of the spatial ID is fine to exceed that of the original token activation, as we subtract the opposing spatial ID to readjust the norm. This is shown in Alg. 2. For confidence intervals, we choose the three layers which had greatest steering effect, and report equivalent layers' effects for the noise case.

## A.6 ID Deviation

**Classifying Model Belief.** For the ID deviation experiment, we classified the model's belief based on its decoded response. If the response contains only the correct spatial relationship (e.g. left) and not the incorrect spatial relationship, it's considered correct. If the response contains only the incorrect spatial relationship, it's considered incorrect. If none or both were present, it's considered nonsense and discarded.

## A.7 Obfuscating Experiments

We take images from COCO_SPATIAL and Gaussian blur different regions, as below. We generate 4 images blurred in incorrect regions in addition to the 1 image with the correct bounding box blurred. For the sensitivity, we take the difference between the outside region which changed the model belief the most, and the bounding box.

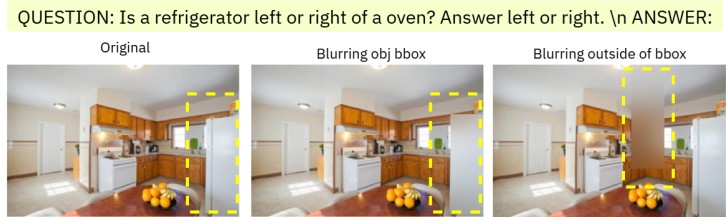

Figure A8: Example of original query and two blurred options. Yellow grid lines are just for visualization.

## A.8 Model Diagnosis Addendum

**Oracle Injection Experiment** To further isolate what model components may be responsible for creating incorrect spatial IDs, we conduct the oracle injection experiment. Specifically, we intervene with the *correct* spatial IDs on the object words at different layers, and see how that changes model accuracy from the control case without any intervention.

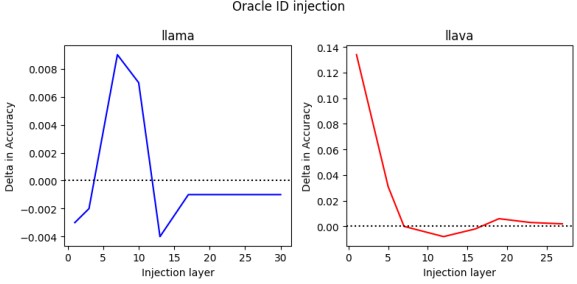

Figure A9: LLaMA and LLaVA evaluation accuracy on synthetic grid-like data with oracle spatial ID injections at varying layers. 0 is baseline model performance, without any intervention.

In accordance with our preliminary conclusion from §4.2, we see that LLaVA models' accuracy increases 13.4% above the baseline when injected with oracle truth spatial IDs at layer 1. This suggests that indeed, if the image encoder had supplied correct spatial information, the downstream LM of LLaVA would have yielded greater accuracy. Intervention on intermediate to later layers in LLaVA has little effect. In LLaMA, we see that intervening on the earliest layers actually has little effect, while intervening on intermediate layers preceding the modality integration layer increases model accuracy by a modest amount ( 1%). Note that the low percentage is likely because LLaMA has higher accuracy on this spatial dataset to begin with. This behavior is in line with our expectation from §4.2, where we do not expect it to benefit greatly from altering image encoder spatial localization performance, but instead benefit from spatial information condensation into the proper tokens.

For this experiment, we evaluated on synthetic images made with objaverse, such as those shown in Fig. A2. The interventions were performed with IDs from layer 17 on LLaMA for all layers including and below 17, and IDs from layer 12 on LLaVA for all layers including and below 12, as these were the layers identified as carrying spatial ID information in these respective models. LLaMA interventions were performed at layers [1, 3, 7, 10, 13, 17, 21, 25, 30, 35] and LLaVA interventions on [1, 5, 7, 12, 16, 19, 23, 27].

## A.9 MODEL FINETUNING WITH SPATIAL LOSS

**Spatial ID Loss Module** In §4.3 we described finetuning Qwen2-2B with a spatial ID augmented loss module. Specifically, we freeze all weights except the MLPs of the last six vision encoder blocks, which we believe are most important for spatial reasoning, and train with synthetic data akin to those shown in Fig. A2. We batch 15 images of the same object but varying locations into a mini-batch, and compute the predicted spatial ID by subtracting the average activation. This is similar to how we extracted the spatial IDs in §2.2.

The validation accuracy and training plots are shown in Fig. A10. We see that with spatial ID loss, model accuracy on the naturalistic validation set (COCO-spatial) increases around 6% (absolute) beyond the baseline plateau, reaching a 90% accuracy in under 2.8k steps.

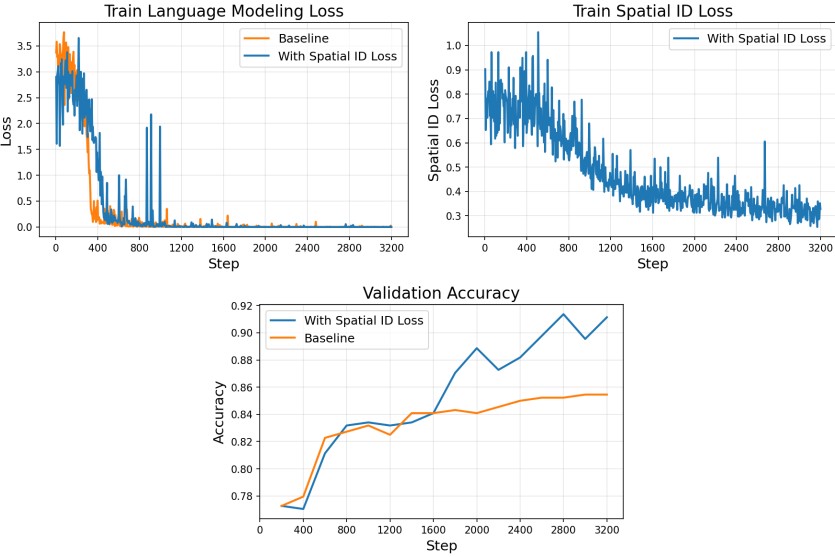

Figure A10: Plots from Qwen2-2B finetuning with and without spatial ID loss

# B EXPERIMENTAL RESULTS ON MORE MODELS

## B.1 SPATIAL GRIDS ON MORE MODELS

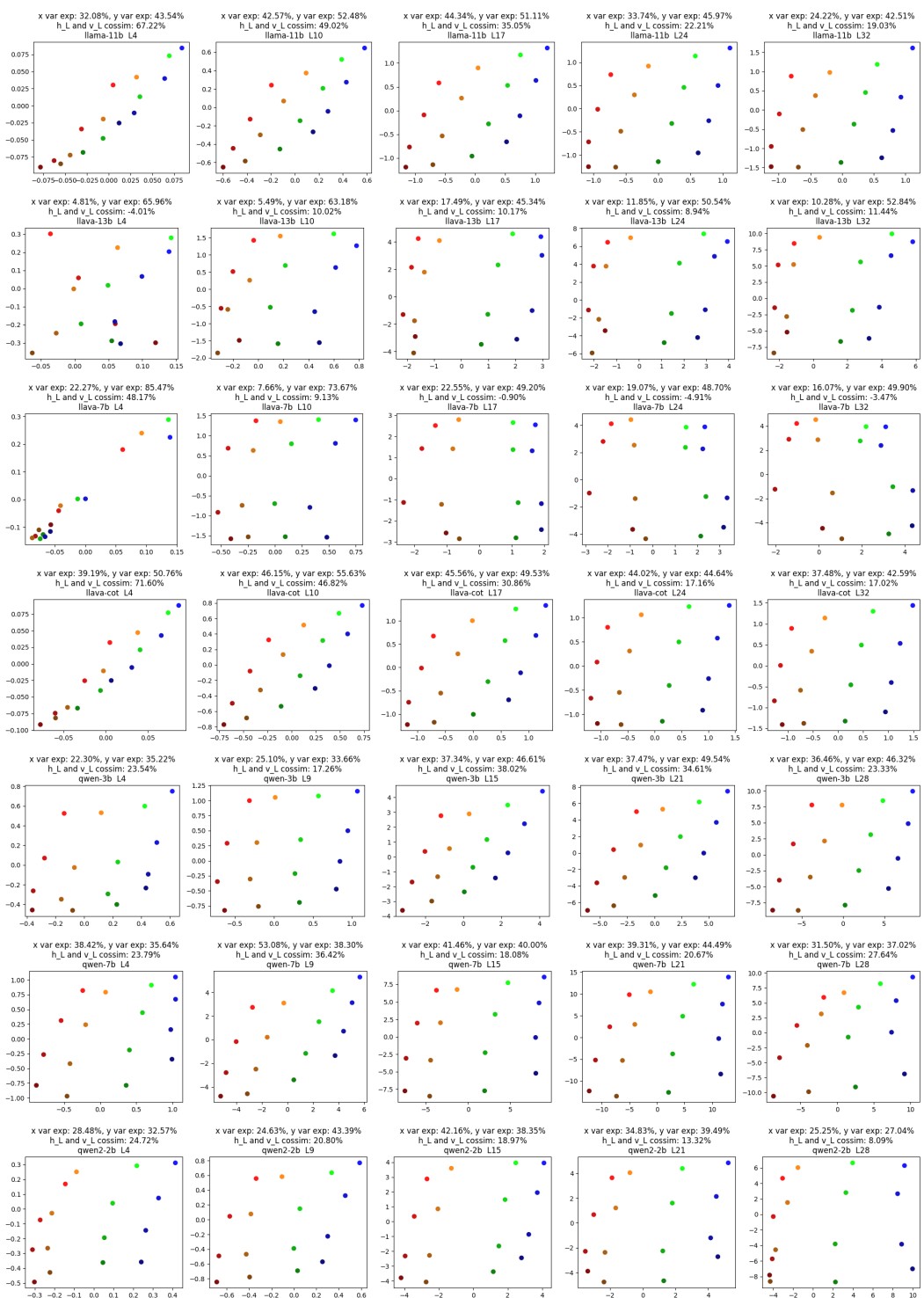

Figure A11: Spatial ID grids for LLaVA, LLaMA, and Qwen models.

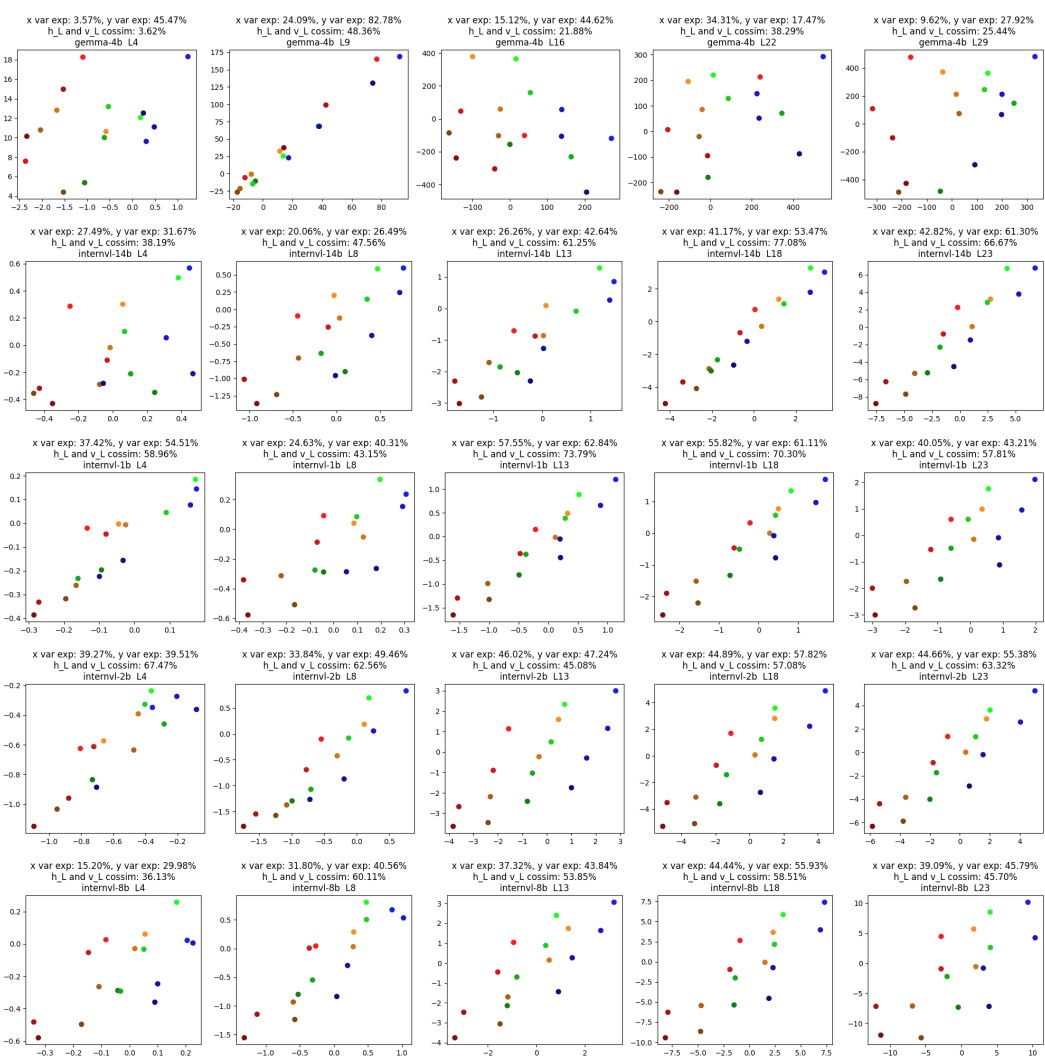

Figure A12: Spatial ID grids for Gemma and InternVL models.

Fig. A11,A12 show spatial ID grids for all models shown in Fig. 2. Subplot headings include % variance explained by each spatial axis, as well as the cosine similarity between the spatial axes. Notably, spatial IDs on some models seem to yield highly correlated $v_L$ and $h_L$, suggesting different spatial directions may be conflated.

## B.2 TEMPORAL GRIDS ON MORE MODELS

Figure A13: Temporal ID grids for VideoLLaMA3, LLaVA-Video, and Qwen2.5.

Across the models, there is a trend for the last frame(s) to be much farther away from the rest of the frames' temporal IDs. This may be a result of the data bias during model training, where a lot of instruction tuning datasets will ask temporal questions that only require paying attention to the last frame (e.g., *did the person leave the room?* only requires looking at the first and last frame, and intermediate nuances are less important).

## C COUNTERFACTUALS

### C.1 SPATIAL IDS FROM NON-OBJECT WORDS

In §2.1 we concluded that spatial information is largely stored in object words at intermediate layers. But could the information storage be spread out across the sequence dimension in internal activations? To test this, we extract spatial IDs from non-object words, per §2.2. Specifically we choose the spatial words in the query format "Is the {obj_word1} {spatial_word1} {spatial_word2} {obj_word2}?".

We then perform steering on object words, as well as non-object words, with both the spatial IDs extracted from object words and non-object words. We use the same steering algorithm as Alg. 2. The results are shown in Table 1. We see that some spatial semantics seems to be extractable from non-object words, and model belief is partially steerable through non-object words when using spatial IDs from object words, likely due to the fact that semantic word meanings are rarely perfectly contained within the initial word token in practical applications. In particular, spatial word tokens are likely to have information bleed over from the object word tokens while performing spatial queries, due to the way attention merges information between similar sequences. Regardless, effects from steering on object words with spatial IDs from object words is by far the strongest.

| Model Name | ID-LR/apply-LR | ID-LR / apply-obj | ID-obj / apply-obj | ID-obj / apply-LR |
|---|---|---|---|---|
| Qwen-3b | 18.77 | 51.19 | **81.23** | 48.46 |
| Qwen-7b | 6.12 | 38.78 | **72.35** | 47.96 |
| LLaVA-7b | 29.83 | 30.51 | **46.26** | 26.19 |
| LLaVA-13b | 8.16 | 19.39 | **48.30** | 15.25 |

Table 1: Spatial IDs extracted from object words and applied onto object words are most successful at steering model beliefs. Spurious effects are observed from IDs extracted from or applied unto unrelated words, but the effects are clearly concentrated on the object words.

## C.2 MIRROR SWAPPING ON NON-OBJECT WORDS

To first showcase on a single sample the difference between mirror swapping on object tokens versus non object tokens, we choose a synthetic example with two objects on a blank background. Fig. A14 shows the results. Here, the green line shows model belief change, and the x axis indicates the layer of intervention. Mirror swapping at just the object words has a slightly less prominent effect than intervening at the object words in addition to immediate neighboring tokens (such as the space preceding the animal word), which captures some of the information bleed. Swapping all tokens except for those belonging to object words, on the other hand, has the smallest observable effect. Hence spatial information is likely concentrated in object tokens.

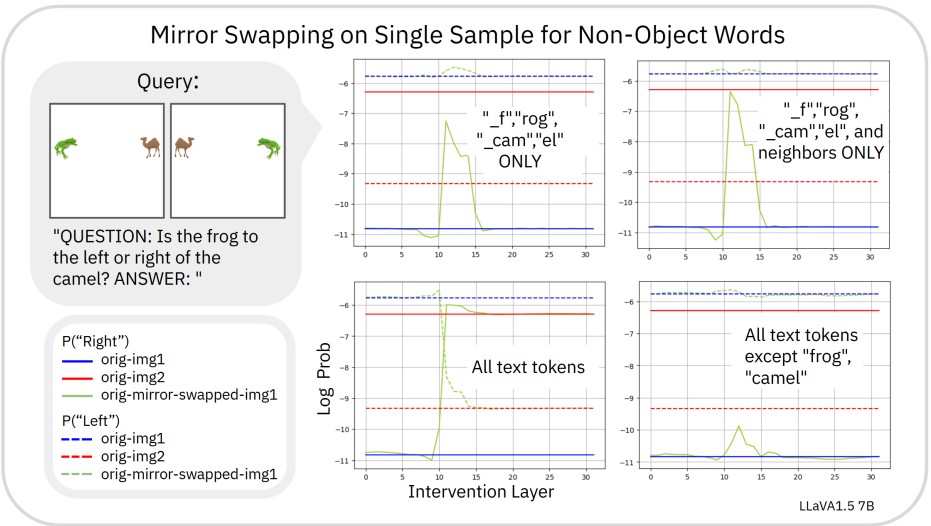

Figure A14: Synthetic image example for mirror swapping. Swapping non-object tokens has minimal impact on model belief change.

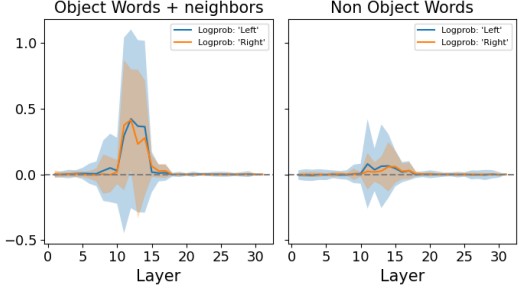

Figure A15: Mirror Swapping non object words

We can now repeat the mirror swapping at non-object tokens at scale on COCO images. Fig. A15 shows the difference between steering on object words and immediate neighboring tokens, versus non object words. Here, the non object words are randomly selected to be the same number of token indices as the object words. We again see that while there is some minor information bleed, the bulk of spatial ID information lies in object word tokens.

### C.3 STEERING EFFECTS ON ORTHOGONAL DIRECTIONS (X VS. Y), (TIME VS. X)

In Fig. 4, we show the results of horizontal steering on "left" vs. "right" beliefs, and vertical steering on "above" and "right". To verify that steering directions can be decoupled, we perform the same steering and observe affects on beliefs of orthogonal directions. We show results of this preliminary analysis on LLaVA. Fig. A16 shows these orthogonal effects. Spatial IDs that are equivalent in the y coordinate but changing in x coordinate do not change beliefs in "above" or "below". Similarly, static x coordinates with a changing y coordinate in spatial IDs has no effect on model belief about "left" and "right".

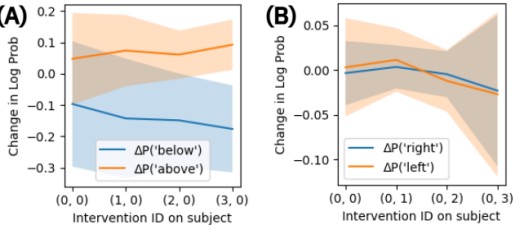

Figure A16: Steering effects of horizontal vectors on vertical beliefs (A) and vertical vectors on horizontal beliefs (B) in LLaVA.

Further, we check orthogonality between the space dimension and temporal dimension in video models. Fig. A17 shows a spatiotemporal ID grid from L11-14 on LLaVA-Video. The IDs are from videos where the object was in one of 8 frames (temporal change), and in one of 3 locations (spatial change). The experimental setup was minimal due to compute limitations. But even in this minimal setting, we see that the spatial and temporal axes are well separated.

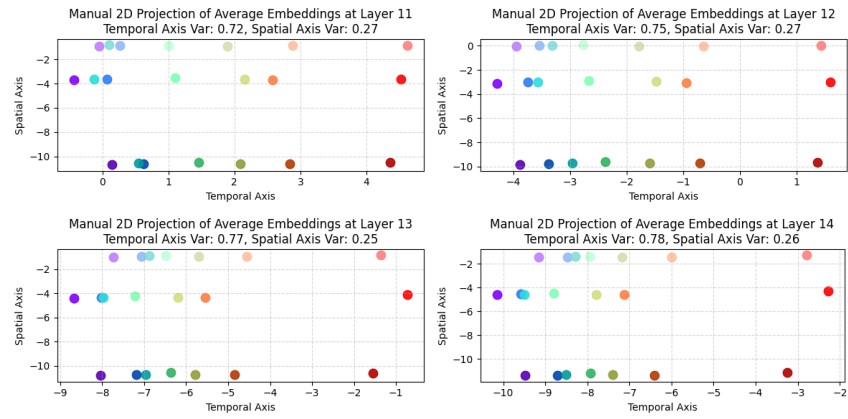

Figure A17: Spatiotemporal ID grid, where y axis is space and x axis is time.

# D ABLATIONS

## D.1 SCALING ANALYSIS FOR SPATIOTEMPORAL ID EXTRACTION

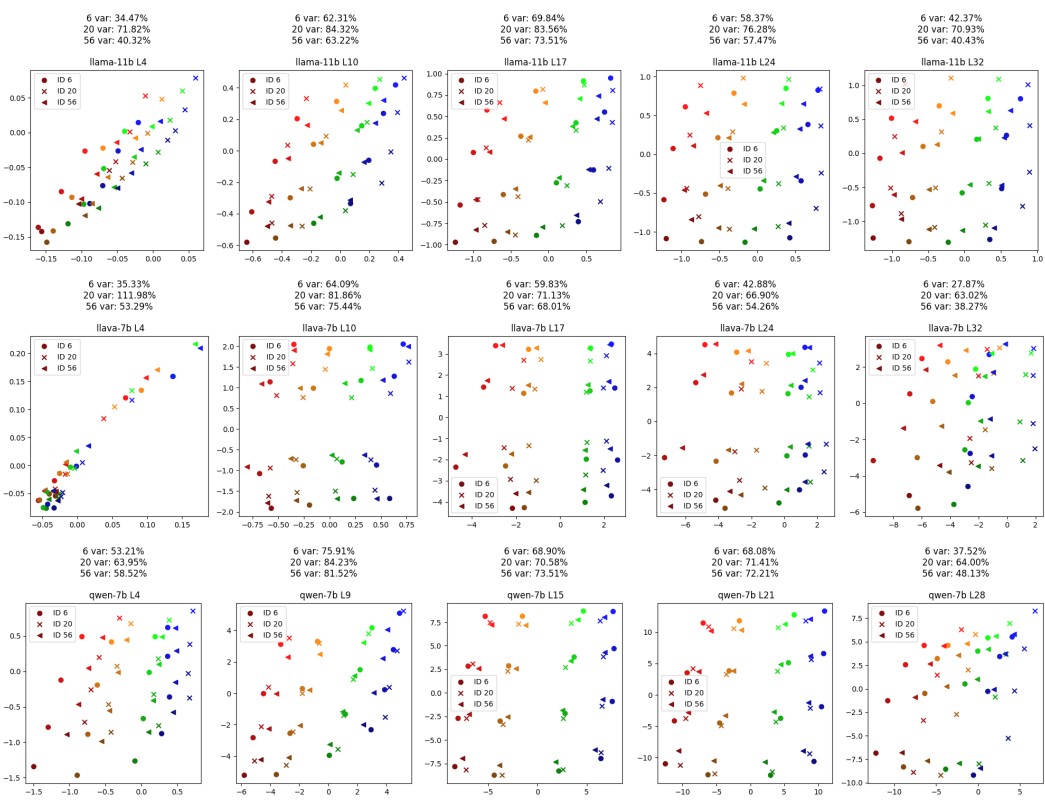

Figure A18: Extracting IDs with 6, 20, and 56 object pair images.

The projection axes are from the 56 object pair case. At intermediate layers, where we expect spatial IDs to be most crucial, we see a tight color-wise clustering, indicating spatial IDs extracted from various numbers of objects still converge. The variance explained by the spatial axes for all spatial ID extraction cases is $\gtrsim 50\%$, showing even at as little as 9 object pairs, we can extract good spatial IDs.

## D.2 VARYING PROMPT WORDING AND OBJECT SIZES DURING EXTRACTION

**Varying prompt wording**. In this work, we use a spatial query in the form "Is the x to the left or right of the y?" to extract spatial IDs from object words. To verify that the choice of prompt does not matter, and that information about spatial location of objects flows into the word activation regardless, we extract spatial IDs from a *plain* prompt in the form "Is there an x or y in the image?". In Fig. A19 we show the results of this extraction. We see that regardless of the input query formatting, spatial information can be extracted from the object words at intermediate.

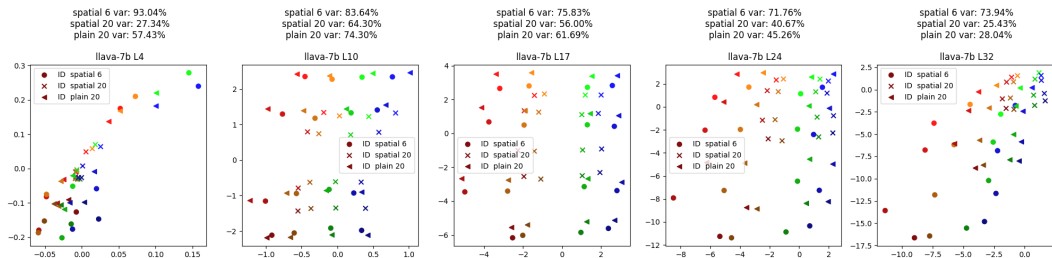

Figure A19: Plain prompts and spatial prompts projected onto spatial axes created from spatial prompts. Colors exhibit tight clustering.

**Varying Image Sizes**. To test that spatial IDs are roughly agnostic to object size, we extract spatial IDs from images where the object is 80px in diameter, 128px, and 176px, then project all extracted spatial IDs onto spatial axes created only from the medium sized object case. The result is shown in Fig A20. While the variance explained by the spatial axes drops by 10~20%, the sptial IDs extracted from different sized objects still exhibit strong in-color clustering, and $\gtrsim 50\%$ of variance are explained by the spatial axes.

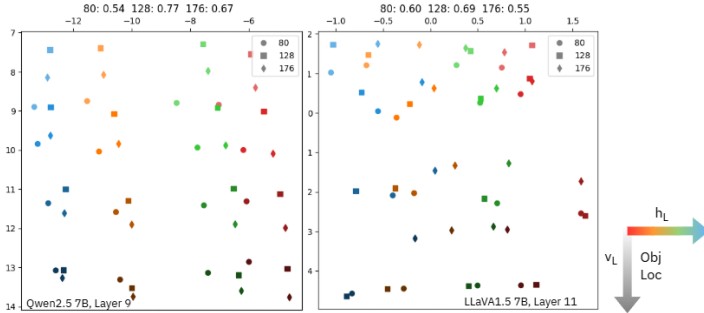

Figure A20: Spatial ID grids for Qwen and LLaVA, extracted from multiple object sizes. Circles are IDs extracted from images where object size was 80px in diameter, square is 128px, and diamond is 176px. On the top row, is the variance explained by the spatial axes for each size case.

## E    THEORETICAL ANALYSIS OF SPATIAL IDS

### E.1    INFORMAL PROOF FOR SPATIAL ID EMERGENCE

**Proposition:** *Universal spatial IDs arises in any VLM using positional encoding, per self attention (Vaswani et al., 2017).*

**Preliminaries**. Consider a VLM layer with one attention head. Let the input sequence contain projected visual tokens $\{x_p\}_{p\in\mathcal{P}}$, where each patch index $p = (i,j)$ lies on an $m \times m$ grid, and text tokens including an object token $o$ (as in prompts "Is there an $o$?"). Define queries, keys, and values $q_o = W_Q r_o$, $k_p = W_K x_p$, $v_p = W_V x_p$, and the standard residual update $r_o \leftarrow r_o + W_{out}\sum_p \alpha_{o\leftarrow p} v_p$ with $\alpha_{o\leftarrow p} = \text{softmax}_p(q_o^\top k_p/\sqrt{d})$.

We make two very weak assumptions.

(1) First, we approximate that each patch vector decomposes as

$$x_p = s_p + P\psi(p) + \varepsilon_p,$$

where $s_p$ encodes content (semantics), $\psi(p) \in \mathbb{R}^{d_\psi}$ is a shared positional basis (e.g., learned 2D embeddings or RoPE-induced features), $P$ maps positional features into model space, and $\varepsilon_p$ is some small deviation. In practice, explicit positional encoding is appended in autoregressive VLMs, so this assumption is explicitly true. In §E.2 we show empirically that positional encodings of VLMs linearly explain spatial IDs.

(2) We also assume that at a patch level, objectness is still encoded such that for images where a visual instance of the object word $o$ occurs at a unique patch $p^* = (i, j)$, the attention kernel is peaked at $p^*$. In other words, $q_o^\top k_{p^*} \gg q_o^\top k_p$ for $p \neq p^*$, so that $\alpha_{o\leftarrow p^*} \approx 1$. Again, this is almost always true in practice, as modality alignment is encouraged during training.

**Proof.** Write the value at patch $p$ using the decomposition:
$$v_p = W_V x_p = W_V s_p + W_V P \psi(p) + W_V \varepsilon_p \tag{12}$$

Under assumption (2), the attention update to the object token is
$$\delta r_o = W_{out} \sum_p \alpha_{o\leftarrow p} v_p \approx W_{out} W_V x_{p^\star} \tag{13}$$

Then we can rewrite Eq.3 as:

$$\Delta_L^{(o)}(p^\star) = r_{o,p^*} - \overline{r_{o,p}}$$
$$= \Big(r_o + W_{out}W_V(s_{p^\star} + P\psi(p^\star) + \epsilon_{p^\star})\Big) - \overline{\Big(r_o + W_{out}W_V(s_p + P\psi(p) + \epsilon_p)\Big)} \tag{14}$$
$$= W_{out}W_V P \Big(s_{p^\star} - s_p + \psi(p^\star) - \psi(p) + \epsilon_p^\star - \epsilon_p\Big)$$

Note that $s_{(o,p^\star)} = s_{(o,p)}$ for any $p$, for the first initial text embedding. Therefore, we can reduce Eq. 14 into:

$$\Delta_L^{(o)}(p^\star) = \Delta_L^{(o)}(i,j) \simeq W_{out}W_V P \Big(\psi(i,j) - \overline{\psi(p)}\Big) \tag{15}$$

This expression is independent of $o$ except through the common matrix $W_{out}U$, so averaging over objects leaves it unchanged. (In practice, we perform the averaging to reduce background noise.)

Notice that $W_{out}W_V P = M$ is fixed for some frozen network, and independent of location. Hence the centered attention update to the object token recovers a fixed linear ID of a shared positional basis, i.e., a universal spatial ID. The implications of the emergence of these intermediate IDs is that a shared spatial vocabulary need only be aligned with their respective positional basis vectors to perform "reasoning". Let $z_o$ be the residual stream at the object token after the update, and let $W_{\mathrm{vocab}}$ be the (approximately linear) readout to logits. Then

$$\ell(\text{LEFT}) - \ell(\text{RIGHT}) \approx (w_{\text{LEFT}} - w_{\text{RIGHT}})^\top \Delta_L(i,j) \approx (w_{\text{LEFT}} - w_{\text{RIGHT}})^\top M \big(\psi(i,j) - \mu_\psi\big) \tag{16}$$

so if $(w_{\text{LEFT}} - w_{\text{RIGHT}})^\top M$ aligns with the $x$-coordinate component of $\psi$, the model correctly predicts spatial words.

**Multi-head and multi-layer accumulation.** For $H$ heads, $M = \sum_{h=1}^H W_{\text{out}}^{(h)} W_V^{(h)} P^{(h)}$; across layers, the contribution composes linearly in the residual stream. The "alignment band" in our experiments corresponds to layers where $||M||$ (or its projection onto the readout) is largest.

### E.2 EMPIRICAL RELATIONSHIP BETWEEN POSITIONAL ENCODING AND SPATIAL IDs

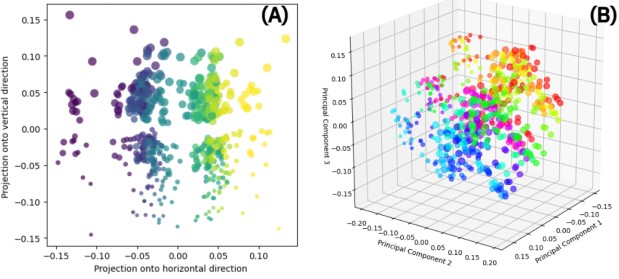

Figure A21: LLaVAPositional Encodings

Fig. A21 shows the patch level positional encodings from LLaVA(which uses the CLIP ViT-L/14 image encoder) projected onto 2 computed spatial axes or 3 principal components. The learned positional encoding vectors clearly have a linear structure, and with reduction in dimension are a linear transformation of the spatial ID grids we extract in §2.2. For a model like Qwen, which starts with fixed Rotary Positional Encodings (RoPE) that are not learned, this separable structure is innate. Previous work has shown that positional encoding in vision encoders continues to be linearly recoverable at penultimate activations (Ren et al., 2023). We are interested in whether this structure is linearly recoverable in a downstream LLM, in the form of spatial IDs, to support §E.1. We show that for the models studied, there exist low rank linear mappings from positional encodings to spatial IDs.

**Setup**. Let $X \in \mathbb{R}^{Nxd}$ be positional encodings for some model and $Y \in \mathbb{R}^{NxM}$ be the spatial IDs extracted. To find their linear relationship, we simply must solve for $W \in \mathbb{R}^{dxM}$ in $Y \approx XW$.

The least–squares solution is obtained with the Moore–Penrose pseudoinverse as $W^{\star} = X^{+}Y$. To impose a rank constraint $r$, we compute the truncated singular value decomposition $X = U\Sigma V^{\top}$ and keep only the top $r$ singular values $\Sigma_r$. Then the rank–$r$ solution is

$$W_r = V_r \Sigma_r^{-1} U_r^{\top} Y \tag{17}$$

The in–sample fit can be quantified by the coefficient of determination:

$$R_r^2 = 1 - \frac{\|Y - XW_r\|_F^2}{\|Y - \bar{Y}\|_F^2}, \tag{18}$$

where $\bar{Y}$ is the column–wise mean of $Y$.

For models like LLaVAand LLaMA, we acquire $X$ by taking the learned positional embeddings. For models that use RoPE (which encodes position through complex rotations applied to query–key pairs) such as Qwen, we need an additional step to extract $X$. Specifically, we can form a RoPE design matrix from the sinusoidal basis functions underlying these rotations and perform the same reduced–rank regression to the extracted spatial IDs $Y$. Each position $p \in \{0, \ldots, N-1\}$ is mapped to sinusoidal features at different frequencies. Let the hidden dimension be $d$, with frequencies

$$\theta_i = 10000^{-\frac{2i}{d}}, \quad i = 0, \ldots, \frac{d}{2} - 1.$$

The RoPE design matrix $X_{RoPE} \in \mathbb{R}^{N \times d}$ is then

$$X_{RoPE}(p) = \left[ \cos(\theta_0 p), \sin(\theta_0 p), \cos(\theta_1 p), \sin(\theta_1 p), \ldots, \cos(\theta_{d/2-1}p), \sin(\theta_{d/2-1}p) \right], \tag{19}$$

with each row of $\Phi$ corresponding to a position $p$.

**Results**. We find that a weight matrix of rank 3 linearly relates the positional encoding matrix to the spatial IDs of a model with $R^2 \geq 0.85$. The three independent weight vectors likely correspond to horizontal, vertical, and radial axes, meaning such structure is preserved in the spatial IDs with high fidelity. Results are shown in Table 2.

| Model | Rank-2 $R^2$ | Rank-3 $R^2$ |
|---|---|---|
| LLaVA1.5-7B | 0.458 | 0.854 |
| LLaMA3.2VL-11B | 0.610 | 0.869 |
| Qwen2.5VL-7B | 0.605 | 0.903 |

Table 2: $R^2$ from low rank $W$

# F LLM USAGE DISCLOSURE

GPT-4 and GPT-5 were used in the process of occasionally coding experiments and editing paper wording.