# OpenReview forum: "Linear Mechanisms for Spatiotemporal Reasoning in Vision Language Models"
_ICLR.cc/2026/Conference — ICLR 2026 Poster_

### Official Review · Reviewer_dAtd · 2025-10-28

**Soundness:** 3
**Presentation:** 3
**Contribution:** 3
**Rating:** 4
**Confidence:** 3

**Summary:**

The paper proposes a linear ID mechanism for spatiotemporal reasoning in VLMs. At intermediate layers, the model encodes an object’s spatial or temporal position as an approximately linear Spatial or Temporal ID within the activation of the corresponding object token, after which most reasoning proceeds in the language channel. The authors also provide an empirical procedure for extracting Spatial IDs. Targeted interventions on these IDs systematically alter the model’s judgments about spatial terms such as left and right, and enabling diagnosis of failure points across architectures, for example vision encoding versus cross modal integration.

**Strengths:**

1.The work unifies the write–then–reason pathway for spatial and temporal information under a linear ID framework, and introduces actionable intervention and diagnosis paradigms, including mirror swapping, arbitrary ID steering, and error attribution for failure cases.

2. The paper reports consistent phenomena and flip rates across 11 VLMs, with noise controls indicating that the effect is not an idiosyncrasy of any single model.

3. The same IDs both explain observed behavior and localize bottlenecks, separating issues in vision encoding, cross-modal binding, and language-stage reasoning, thereby offering concrete guidance for model engineering.

4. The study reproduces the same linearly steerable structure in video models as Temporal IDs, aligning interventions with before and after relations and suggesting a more general mechanism.

**Weaknesses:**

1. The Mirror Swapping design localizes key bottlenecks and supports the claim that Spatial IDs are bound to object tokens by replacing only the object tokens. However, it omits a crucial control condition that would replace non-object tokens under an otherwise identical setup, which is necessary to more directly test whether the observed effects indeed hinge on object token binding rather than on broader sequence perturbations.

2. The causal attribution in §4.2 across the stages of vision encoding, cross modal binding, and language readout is not fully convincing. The inference that Spatial ID errors imply the language stage is not the bottleneck lacks necessity level causal testing; in other words, a deviation in the Spatial ID does not rule out the possibility that the language decoder exists erroneous priors during readout, a phenomenon that has been reported as language side priors or language bias in VQA tasks[1][2]. In addition, using sensitivity to occluding the ground truth box to distinguish vision encoding from cross modal binding conflates misidentification or localization failure with miswriting into the object token, making it difficult to uniquely localize the failing component. Overall, the current Spatial ID instrumentation does not cleanly isolate the faulty segment of the architecture.

3. The paper largely remains at the level of revealing and validating the Spatial ID mechanism, without translating this mechanism into concrete corrective or editing procedures for existing models and without empirical evaluation of such interventions in a remediation setting.

[1] Ramakrishnan, Sainandan, Aishwarya Agrawal, and Stefan Lee. "Overcoming language priors in visual question answering with adversarial regularization." Advances in neural information processing systems 31 (2018).
[2] Leng, Sicong, et al. "Mitigating object hallucinations in large vision-language models through visual contrastive decoding." Proceedings of the IEEE/CVF Conference on Computer Vision and Pattern Recognition. 2024.

**Questions:**

NA

---

> ### Author Response · Authors · 2025-11-20
> **Part 1/2**
>
> Thank you for your detailed feedback! We are glad that you find our work to be "introducing actionable intervention and diagnosis paradigms" and to be consistently showing that our proposed mechanism is "not an idiosyncrasy of any single model". Below we discuss the proposed experiments and comments:
>
>
>     Mirror swapping… omits a crucial control condition that would replace non-object tokens under an otherwise identical setup.
>
> This is a great point, adding such a control condition would certainly strengthen our claim that spatial IDs represent object-level localization. We have updated our manuscript to include a single-sample example (**Fig. A14**) as well as a population level mirror swapping result (**Fig. A15**) showing that swapping non-object words has significantly lower influence on model belief than object tokens and its immediate neighbor indices. Below is a summary table highlighting the belief changes resultant from object word swapping or non object word swapping on COCO-spatial:
>
> | layer | all object words + neighbors (%)| non object words (%) |
> |-------|----------------------------------------|-------------------|
> | 1     | -0.06                                  | -0.27             |
> | 5     | 0.11                                   | 0.01              |
> | 10    | **33.61**                                | **5.20**              |
> | 15    | 1.84                                   | 2.06              |
> | 20    | 0.04                                   | 0.02              |
> | 25    | 0.07                                   | -0.13             |
> | 30    | 0.00                                   | 0.00              |
>
>
>
> In summary, we can see that at intermediate layers, the belief change effect from steering is greatest for the object word tokens plus its immediate neighbors. The change in belief from swapping all text tokens sans the object words is much smaller in comparison, by an order of magnitude. The fact that it is nonzero indicates there is some information bleed, but as swapping on object word tokens has 500+% greater influence over the model belief, we can conlcude that spatial information is concentrated on object word tokens.
>
>
>     The inference that Spatial ID errors imply the language stage is not the bottleneck lacks necessity level causal testing; in other words, a deviation in the Spatial ID does not rule out the possibility that the language decoder exists erroneous priors during readout, a phenomenon that has been reported as language side priors or language bias in VQA tasks.
>
> This is a great nuanced point. If the language stage were the primary bottleneck, interventions on spatial IDs alone would not consistently shift model beliefs. However, we observe robust steering effects with injected spatial IDs. **Fig. 2** in our original paper, as well as **Fig. 9**, which we added for the rebuttal, showcases that strong models are more than 50% faithful to spatial ID interventions. Below is a summary table for **Fig. 9**, which shows how the "steerability" of models correlate to their accuracy on a spatial reasoning benchmark:
>
> | Model | Steerability (mean ± std) | Accuracy on COCO-spatial |
> |-------|----------------|---------|
> | gemma-4b | 0.15 ± 0.045 | 0.72 |
> | llava-13b | 0.062 ± 0.1 | 0.78 |
> | gemma-12b | 0.2 ± 0.063 | 0.82 |
> | llava-7b | 0.16 ± 0.12 | 0.84 |
> | qwen2-2b | 0.37 ± 0.076 | 0.87 |
> | llama-11b | 0.35 ± 0.13 | 0.88 |
> | internvl-1b | 0.39 ± 0.14 | 0.91 |
> | qwen-3b | 0.58 ± 0.11 | 0.92 |
> | internvl-2b | 0.53 ± 0.11 | 0.94 |
> | internvl-8b | 0.43 ± 0.13 | 0.94 |
> | internvl-14b | 0.56 ± 0.12 | 0.95 |
> | qwen-7b | 0.48 ± 0.14 | 0.96 |
>
> The table has been sorted by increasing accuracy on COCO-spatial. We see that on the whole, models that have stronger spatial IDs, and are therefore steered more readily, have higher accuracy. The linear regression on Steerability vs. Spatial Evaluation Accuracy on these models has a slope of 0.3826, with an R^2 of 0.8018.
> For this analysis, we compute the “steerability” as the difference between the change of belief resultant from steering with opposing spatial IDs, minus the change of belief from steering with noise.
>
> The fact that spatial IDs directly change model beliefs 50% of the time in large models indicates that the language module is capable of faithfully propagating spatial encodings to a large degree, and understanding how spatial IDs contribute to failure modes is thus meaningful.
>
> Of course, we see throughout the paper that even with the correct spatial IDs, model beliefs are not always perectly steered to the correct answer. In fact, if we are steering model beliefs 50% of the time with spatial IDs, then the rest of the time the language model is indeed the likely failing point.
> We have updated Section 4 to be more precise with our wording, and acknowledge that language model failures beyond spatial IDs cannot be disregarded as a result of spatial ID-based diagnosis, citing the references you provided.

---

> ### Author Response · Authors · 2025-11-20
> **Part 2/2**
>
> (continued)
>
>        In addition, using sensitivity to occluding the ground truth box to distinguish vision encoding from cross modal binding conflates misidentification or localization failure with miswriting into the object token.
>
> We see your point! With a clearer signal showing that poor spatial IDs in LLaVA are correlated to poor object localization, our diagnosis method will be stronger.
> To ascertain if that is the case, we perform an *oracle injection* experiment. Specifically, we intervene with the *correct* spatial IDs on the object words at different layers, and see how that changes model accuracy from the control case without any intervention. If LLaVA's faulty object recognition was the culprit behind its poor spatial IDs, we would see that intervening at early LM layers would improve model performance in LLaVA, but less so in LLaMA.
> A more detailed figure has been added in **Fig A9** in the supplements. A summary table is below:
>
> *Oracle injection on LLaVA*
> | Injection layer | Δ Accuracy LLaVA (%)| Δ Accuracy LLaMa (%) |
> |-----------------|-----------|---------|
> | 1 | **+13.40**| -0.30 |
> | 7 | +00.00 |**+0.90** |
> | 12/13 | -0.80 | -0.40 |
> | 16/17 | -0.20 | -0.10 |
> | 27/30 | +0.20 | -0.10 |
>
> In accordance with our preliminary conclusion from **Sec. 4.2**, we see that LLaVA models' accuracy increases 13.4% above the baseline when injected with oracle truth spatial IDs at layer 1. This suggests that indeed, if the image encoder had supplied correct spatial information, the downstream LM of LLaVA would have yielded greater accuracy. Intervention on intermediate to later layers in LLaVA has little effect. In LLaMA, we see that intervening on the earliest layers actually has little effect, while intervening on intermediate layers preceding the modality integration layer increases model accuracy by a modest amount (~1\%). Note that the low percentage is in part because LLaMA has higher accuracy on this spatial dataset to begin with. This behavior is in line with our expectation from **Sec. 4.2**, where we do not expect LLaMA to benefit greatly from altering image encoder spatial localization performance, but instead benefit from spatial information condensation into the proper object tokens. Small negative deviations in accuracy from oracle intervention in other layers is likely a result of the spatial ID injection causing activations to be slightly out of distribution.
>
>
>     Paper largely remains at the level of revealing and validating the Spatial ID mechanism, without translating this mechanism into concrete corrective or editing procedures for existing models and without empirical evaluation of such interventions.
>
> We agree that incorporating spatial IDs into improving model performance will strengthen our argument for the importance of this mechanism! We demonstrate how spatial IDs can help model performance through a simple finetuning experiment.
>
> More specifically, we finetune Qwen2-2B on a synthetic dataset similar to the one used to extract spatial IDs, and evaluate on COCO-Spatial. For the spatially informed training, we introduce an additional loss module at layer 11 that computes the cosine similarity between the predicted and ground-truth spatial ID at that layer. This spatial ID loss is added to the standard language modeling objective, providing extra supervision. We also perform a control training where we only use the stanard objective. We revised the manuscript to provide more precise experimental details for this finetuning in **Sec. 4.3** and **Sec. A9**. In summary, we find that indeed, incorporating spatial ID alignment into the finetuning loss allows the model to generalize to the naturalistic validation set quicker.
>
>
> Below is the finetuning result from the control case using just the simple LM loss:
>
> |     Num Steps              | 0    | 800  | 1600  | 2400  | 3200  |
> |-------------------|------|------|-------|-------|-------|
> | LM Loss           | 3.3  | 0.05 | <0.01 | <0.01 | <0.01 |
> | COCO Val Accuracy | 0.77 | 0.83 | 0.84  | 0.85  | **0.85**  |
>
> And below is the result for the combined LM and spatial ID loss:
>
> |          Num Steps         | 0    | 800  | 1600  | 2400  | 3200  |
> |-------------------|------|------|-------|-------|-------|
> | LM Loss           | 2.71 | 0.04 | <0.01 | <0.01 | <0.01 |
> | Spatial ID Loss   | 0.75 | 0.58 | 0.41  | 0.36  | 0.33  |
> | COCO Val Accuracy | 0.77 | 0.83 | 0.84  | 0.88  | **0.91**  |
>
> With the spatial ID loss, the model attains above 90\% accuracy on COCO-spatial at 3.2k steps, which is a 6% absolute advantage over the control case.
> We have added training plots to **Fig. A10** in the supplements.
>
> Thank you again for your thorough review, and for many helpful suggestions that improved our work!

---

> > ### Author Response · Authors · 2025-11-27
> > **Gentle Reminder of Rebuttals**
> >
> > Thank you again for taking the time to review our paper. We would like to gently remind you that there is just a week left before the rebuttal period deadline. To address your concerns, we have added new results from 1. *fine-grained mirror swapping controls to ascertain that spatial IDs are embedded in object word activations*, as well as results showing that 2. *stronger models show strong spatial ID based steering*, which suggests their downstream LMs were faithful to earlier layer spatial IDs, results showing 3. *oracle spatial ID injection in layer 1 of the LM helps LLaVA performance but not LLaMA*, strengthening our diagnosis conclusion that LLaVA's image encoder is its bottleneck, and some final additions showing that 4. *using spatial-ID guided loss module promotes faster generalization during fine-tuning*. We believe these new additions demonstrate that spatial IDs mediate a complex array of spatial relationships, and that they are useful for improving model performance.
> >
> > We would greatly value any further feedback or confirmation of whether we have addressed your comments sufficiently.
> > Thank you, and happy almost holidays!

---

### Official Review · Reviewer_4Gni · 2025-10-30

**Soundness:** 3
**Presentation:** 3
**Contribution:** 3
**Rating:** 6
**Confidence:** 2

**Summary:**

This paper investigates how and where VLMs combine spatial visual information with textual representations to facilitate capabilities such as spatiotemporal reasoning.  The authors derive and identify spatial IDs, which encode object locations in an $m\times m$ grid.  Spatial IDs are then used for interventions such as steering, as well as a variety of insightful analyses such as the lack of depth representation in VLMs.  The paper also investigates Temporal IDs, rounding out their investigation of spatiotemporal reasoning in VLMS.

**Strengths:**

The strengths of the work include its discovery and derivation of spatial IDs, and its thorough utilization for a variety of insights.  The paper defended spatial IDs thoroughly, through adversarial steering as well as how deviations from the ground truth spatial ID results in worse predictions.

**Weaknesses:**

Whereas the Spatial ID was thoroughly explored, much less attention was put on Temporal IDs.  Furthermore, the "before" and "after" evaluation is a bit more simplistic than spatial reasoning.  One expects a smoother interpolation; but in Figure 9(C) for 'After', both changes in logprob decrease past frame 5.

Furthermore, the queries are quite limited to simple position-based analysis.  General reasoning should utilize other attributes beyond position, such as properties of the model (e.g. an ice cream in sunlight should most likely melt - reasoning about this change over time requires less positional information).  For such settings, it is not clear how the current approach can provide insight into how the model is performing its reasoning.  It would be interesting to see if object-specific attributes can also be encoded within or alongside spatial IDs for more complete isolation of where the binding with text is performed; or if the authors can provide insight into where such other attributes may be found and where they are bound.

**Questions:**

For the mean embedding of object $o$, is this computed across a batch of examples of the particular object?  For example is it averaged across many different apple photos (positioned in different ($i, j$) per image)?

Is the $(i, j)$ spatial coordinate for an object just one singular coordinate?  If the object takes up a lot of space in the image you can imagine it encompassing many coordinates in the $m \times m$ grid.  In such a case, is only the centroid of the object used as the spatial position?  What about objects that span many discretized squares where the centroid may not actually be the position of the object (e.g. a bike; the centroid may be empty air)?

---

> ### Author Response · Authors · 2025-11-20
> **Part 1/2**
>
> Thank you for your thoughtful comments and questions! We are happy to hear that you found our work to have "a variety of insightful analyses", and that you felt our "paper defended spatial IDs thoroughly"! We address the suggestions and questions below:
>
>     Whereas the Spatial ID was thoroughly explored, much less attention was put on Temporal IDs… "before" and "after" evaluation is a bit more simplistic...
>
> Indeed, the temporal IDs section was a lot shorter than the rest of the sections due to space limitations. Our supplements contain richer exploration of temporal IDs and their coexistence with spatial ID. For example, in **Fig. A16**, we show that temporal IDs are cleanly orthogonal from spatial IDs for intermediate layers of LLaVA-Video, alluding to independent temporal and spatial mechanisms at play.
>
>
> We also glean some insights on data biases that seem to exist in temporal IDs, akin to Sec. 4.1. for spatial IDs. In **Fig. A13** we show extracted temporal IDs from three distinct model types. Notably, temporal IDs exhibit a strong bias towards the first and last frames in a continuous scene, while intermediate frames are less well articulated. This bias is least noticeable in LLaVA-Video, the strongest video model we tested. Our newly added **Fig. 9** illustrates that stronger spatial IDs correlate to stronger models overall, and that is likely the case for temporal IDs and video models as well.
>
> Overall, we certainly focused on spatial IDs for this paper for most of our analyses for the sake of space and focus, but we hope you find the supplementary results insightful. We hope to continue this line of investigation, and encourage others to further explore, towards richer temporal ID understanding in video models!
>
>
>     Queries are quite limited to simple position-based analysis… interesting to see if object-specific attributes can also be encoded within or alongside spatial IDs for more complete isolation of where the binding with text is performed.
>
> Thank you for this interesting suggestion! To your point about visual reasoning about object-specific attributes beyond direct spatial queries, we found out that *object-specific attribute binding* also seems to be mediated by spatial IDs. More specifically - when answering a question about what attribute a certain object has, the model seems to match the spatial ID of the object with all attributes present in the image, to find the best matching pair.
>
> We were able to reach this conclusion after we first designed and conducted mirror swapping experiments for a "color binding" visual reasoning scenario. We have included new results in the revised version of our manuscript.
>
> The experimental setup has two original images, each with two objects, and our query to the model is a non-spatial query like "What color is the *object*?". This time, we keep the objects in the same location, but change their colors - so the first image could contain a red backpack and blue gloves, while the second image contains red gloves and a blue backpack. We find that mirror swapping the activations of color words (e.g. *red* and *blue*) reliably changes the model's belief about what color an object is.
>
> We add new results from a single sample mirror swap (Fig **A6**) as well as averaged over many samples (Fig **A7**) for "color binding" in the supplements of our modified submission. Below we show a summary table of the color binding experiments:
>
>
> | layer | all image patches (%)| all text tokens  (%)| color tokens only (%) | non-color tokens (%) |
> |-------|-------------------|------------------|--------------------|------------------|
> | 1     | 97.92             | 0.88             | 0.57               | 1.40             |
> | 5     | **98.22**         | 0.13             | 0.23               | 0.63             |
> | 10    | 43.05             | 49.80            | **34.29**          | 1.13             |
> | 15    | 20.15             | 82.86            | 19.66              | 2.58             |
> | 20    | 2.50              | 96.83            | 8.84               | 4.23             |
> | 25    | 0.53              | 98.90            | 0.15               | 3.18             |
> | 30    | 0.00              | **99.37**        | 0.00               | 0.00             |
>
>
> As with the spatial mirror swapping, we see that image patches cause the highest change in model belief in early layers, while swapping all text tokens has greater effects in later layers. Swapping color words around layer 10 results in a 34% model belief change, while swapping non-color tokens at the same layer has minimal effect (~1%).
> This suggests that VLMs answer such visual reasoning queries by matching the attribute token which is most spatially similar to the question subject. In addition to reasoning about attribute binding, spatial IDs may mediate many other types of visual reasoning beyond direct spatial queries.

---

> > ### Author Response · Authors · 2025-11-20
> > **Part 2/2**
> >
> > **Questions:**
> >
> >     “Is this computed across a batch of examples of the particular object?”
> >
> > Yes, for each reference-object pair, we synthetically generate a batch of 15 images where an object (appears once in the image) is varied between 15 grid locations (i in (0,4),j in (0,4)) minus the grid that contains the fixed reference. We then compute the average activation across the batch to acquire the spatial ID. **Fig. A4** in the Supplements shows a more precise extraction pipeline diagram.
> >
> >     “Is the spatial coordinate for an object just one singular coordinate? If the object takes up a lot of space in the image you can imagine it encompassing many coordinates in the grid. In such a case, is only the centroid of the object used as the spatial position?...”
> >
> > Great question! We actually discuss this in **Fig A19** in the supplements (which are very long, so we do not blame you for missing it!). We extracted spatial IDs from images where the objects were different sizes (specifically, 80px, 128px, and 176px on a 1040px square). There is certainly some size encoding difference, as can be seen by the fact that the PCs explain ~10% less variance in spatial IDs from images that contained objects of varying sizes, but for all object sizes more than 50% of the internal activations' variance can be explained by their spatial localization. As such, generally we see that the centroid of the object, regardless of its size, seems to be representative of its location in the form of spatial IDs within the model internals. Size is likely encoded in a separate dimension.
> >
> > Thank you again for your interesting suggestions and questions!

---

> > > ### Author Response · Authors · 2025-11-27
> > > **Gentle Reminder of Rebuttals**
> > >
> > > Thank you again for taking the time to review our paper. We would like to gently remind you that there is just a week left before the rebuttal period deadline. To address your concerns, we have highlighted *further exploration of temporal IDs*, demonstrated that *object-specific attribute based reasoning is also mediated by spatial IDs*, and provided clarifications for the spatial ID extraction process. We believe these new additions and explanations show the wide-spread applications of spatial IDs.
> > >
> > > We would greatly value any further feedback or confirmation of whether we have addressed your comments sufficiently.
> > > Thank you, and happy almost holidays!

---

### Official Review · Reviewer_zD2N · 2025-11-01

**Soundness:** 3
**Presentation:** 3
**Contribution:** 3
**Rating:** 6
**Confidence:** 3

**Summary:**

This paper identifies that VLMs encode spatial relationships through linear features called "spatial IDs" bound to object token representations. Through extraction and causal intervention experiments, the authors demonstrate that these spatial IDs can be manipulated via representation steering to control model outputs on spatial reasoning tasks. The analysis extends to video models, revealing similar linear temporal ID mechanisms for temporal reasoning.

**Strengths:**

1. Interesting mechanistic discovery: Identifying spatial IDs as linear features that mediate spatial reasoning in VLMs is novel and insightful, revealing how models bind location information to object tokens for subsequent linguistic processing.
2. Robust experimental validation: The paper provides rigorous evidence through well-controlled experiments (mirror-swapping with controls, causal interventions across 11 models achieving 64.4% vs 29.5% belief swap rates) and validates the framework across multiple datasets and model types.

**Weaknesses:**

1. Limited spatial relation coverage: The analysis focuses primarily on simple binary spatial queries ("left/right", "up/down"), while more complex and diverse spatial relationships like "near", "far", "between", or "surrounded by" remain unexplored. The generalizability of the linear spatial ID framework to these richer spatial concepts is unclear, limiting the scope of the findings.
2. Insufficient guidance for model improvement: While the paper offers valuable diagnostic insights, it lacks actionable strategies for improving VLM training or architecture. The identified issues (e.g., depth-height conflation in Section 4.1) are not accompanied by proposed solutions or experimental validation of potential fixes. Demonstrating how spatial ID insights can inform better training objectives, architectural modifications, or data curation would significantly enhance the practical impact of this work.

**Questions:**

What causes the depth-vertical conflation? Section 4.1 identifies depth-height conflation but does not determine whether this stems from training data biases (e.g., perspective projection correlating height with distance), architectural limitations of 2D vision encoders, or the lack of 3D positional encodings. Without understanding the cause, no solutions are proposed or tested. Given that depth reasoning is critical for real-world applications, this represents a significant missed opportunity to translate mechanistic insights into actionable improvements such as 3D spatial IDs, depth-aware training objectives, or multi-view architectures.

---

> ### Author Response · Authors · 2025-11-20
> **Part 1/2**
>
> Thank you for your helpful comments and questions! We are grateful that you found our work to be an "interesting mechanistic discovery" that is "novel and insightful", with "robust experimental validation". We address the proposals and questions you raised below:
>
>
>     Limited spatial relation coverage: ... “more complex and diverse spatial relationships like "near", "far", "between", or "surrounded by" remain unexplored”
>
>
> This is a great point. Exploring whether spatial IDs mediate varied spatial relationships beyond those considered in the main paper will certainly extend the scope of our findings, and make a stronger case that we have identified a crucial and ubiquitous mechanism for spatial reasoning in VLMs.
>
> In our revised submission, we have updated **Fig. 6** with results of spatial ID steering for a few more types of spatial relationships, as suggested. Spatial IDs indeed mediate reasoning for relative distance (e.g. near/far), as well as the three-way relationship between multiple things (e.g. A in between B and C).
>
> In addition to the figure, below is a summary table:
>
> *Layer 12 — Original: object @ left*
> | Intervention | mean ΔP('near') | mean ΔP('far') | far - near |
> |-------------|------------------|------------------|------------|
> | (0, 0) | +0.1040 | -0.3259 | -0.4299 |
> | (1, 0) | +0.0768 | -0.1227 | -0.1996 |
> | (2, 0) | -0.0393 | +0.0355 | +0.0748 |
> | (3, 0) | -0.0596 | +0.0163 | +0.0759 |
>
> *Layer 12 — Original: object @ right*
> | Intervention | mean ΔP('near') | mean ΔP('far') | far - near |
> |-------------|------------------|------------------|------------|
> | (0, 0) | -0.3528 | +0.0560 | +0.4088 |
> | (1, 0) | -0.1467 | -0.0000 | +0.1467 |
> | (2, 0) | +0.1660 | +0.0609 | -0.1051 |
> | (3, 0) | +0.2498 | +0.0927 | -0.1571 |
>
>
> More specifically, we find that when the object is to the left, altering the spatial ID of the subject towards the right increases the likelihood of "far" and decreases that of "near", and vice versa if the object is to the right.
>
>
> Here is a brief summary table for in-betweenness:
>
> *Original: in-between (L13)*
> | Intervention | mean ΔP('yes' = in-between) | mean ΔP('no' = not in-between) | yes - no |
> |-------------|------------------|------------------|------------|
> | (2, 0) | -0.0031 | -0.0041 | -0.0010 |
> | (2, 1) | -0.0010 | +0.0073 | +0.0083 |
> | (2, 2) | +0.0046 | -0.0388 | -0.0434 |
> | (2, 3) | +0.0280 | -0.1696 | -0.1975 |
>
> *Original: not in-between (L13)*
> | Intervention | mean ΔP('yes' = in-between) | mean ΔP('no' = not in-between) | yes - no |
> |-------------|------------------|------------------|------------|
> | (2, 0) | -0.0056 | +0.0276 | +0.0332 |
> | (2, 1) | -0.0023 | +0.0180 | +0.0203 |
> | (2, 2) | +0.0026 | -0.0205 | -0.0230 |
> | (2, 3) | +0.0140 | -0.0767 | -0.0907 |
>
> Similarly, we find that bringing a subject closer to be surrounded by two objects increases the model's belief that the subject is *in between* those objects. Thank you for your suggestion to perform these additional analyses!

---

> ### Author Response · Authors · 2025-11-20
> **Part 2/2**
>
> (continued)
>
>       Demonstrating how spatial ID insights can inform better training objectives, architectural modifications, or data curation would significantly enhance the practical impact of this work.
>
> Thank you for this pointer.
> Incorporating spatial IDs to improve benchmark performance of VLMs will certainly strengthen our paper, and highlight the importance of the spatial ID mechanism. We demonstrate how spatial IDs can help model performance through a simple finetuning experiment.
>
> More specifically, we finetune Qwen2-2B on a synthetic dataset similar to the one used to extract spatial IDs, and evaluate on COCO-Spatial. For the spatially informed training, we introduce an additional loss module at layer 11 that computes the cosine similarity between the predicted and ground-truth spatial ID at that layer. This spatial ID loss is added to the standard language modeling objective, providing extra supervision. We also perform a control training where we only use the stanard objective. We revised the manuscript to provide more precise experimental details for this finetuning in **Sec. 4.3** and **Sec. A9**. In summary, we find that indeed, incorporating spatial ID alignment into the finetuning loss allows the model to generalize to the naturalistic validation set quicker.
>
>
> Below is the finetuning result from the control case using just the simple LM loss:
>
> |     Num Steps              | 0    | 800  | 1600  | 2400  | 3200  |
> |-------------------|------|------|-------|-------|-------|
> | LM Loss           | 3.3  | 0.05 | <0.01 | <0.01 | <0.01 |
> | COCO Val Accuracy | 0.77 | 0.83 | 0.84  | 0.85  | **0.85**  |
>
> And below is the result for the combined LM and spatial ID loss:
>
> |          Num Steps         | 0    | 800  | 1600  | 2400  | 3200  |
> |-------------------|------|------|-------|-------|-------|
> | LM Loss           | 2.71 | 0.04 | <0.01 | <0.01 | <0.01 |
> | Spatial ID Loss   | 0.75 | 0.58 | 0.41  | 0.36  | 0.33  |
> | COCO Val Accuracy | 0.77 | 0.83 | 0.84  | 0.88  | **0.91**  |
>
> With the spatial ID loss, the model attains above 90\% accuracy on COCO-spatial at 3.2k steps, which is a 6% absolute advantage over the control case.
> We have added training plots to **Fig. A10** in the supplements.
>
> Both suggestions you mentioned yielded clear and actionable updates which improved our paper, thank you!
>
> **Questions:**
>
>     What causes the depth-vertical conflation?
>
> Great question. In reality, it is most likely a multitude of factors at play, but the most likely culprit is the data contamination.
> In [1],  the authors do a corpus study on LAION2B and find that “common spatial prepositions occur in less than 0.2% of the training data”, and that “when they do occur, they can be ambiguous or extraneous to the image, e.g., “left” defined from the viewer’s perspective vs the subject’s”.
>
> In fact, in 100% of the evaluation examples in WhatsUP, a spatial evaluation benchmark, the ground truth location for object A when A is "behind" B is always that A has a higher y-coordinate, and vice versa for "in front of". This is an understandable conflation, since in most naturalistic settings when we take pictures where A is "behind" B, for A to not be occluded it will likely "above" B as well. Nonetheless, a fully generalizable LVLM should be able to handle edge cases with complex 3D relationships. It seems like more varied training data, or built in structured 3D priors, can help overcome these 2D-based depth priors.
>
>
> [1] Amita Kamath, Jack Hessel, and Kai-Wei Chang. What’s” up” with vision-language models? investigating their struggle with spatial reasoning. arXiv preprint arXiv:2310.19785, 2023.

---

> > ### Author Response · Authors · 2025-11-27
> > **Gentle Reminder of Rebuttals**
> >
> > Thank you again for taking the time to review our paper. We would like to gently remind you that there is just a week left before the rebuttal period deadline. To address your comments, we have added new analyses incorporating *additional spatial relationships beyond above/below/left/right*, and shown that *using spatial-ID guided loss module promotes faster generalization during fine-tuning*. We believe these new additions demonstrate that spatial IDs mediate a complex array of spatial relationships, and that they are useful for improving model performance.
> >
> > We would greatly value any further feedback or confirmation of whether we have addressed your comments sufficiently.
> > Thank you, and happy almost holidays!

---

### Official Review · Reviewer_rYNZ · 2025-11-09

**Soundness:** 4
**Presentation:** 4
**Contribution:** 3
**Rating:** 4
**Confidence:** 3

**Summary:**

The paper studies how VLMs do spatial/temporal reasoning by identifying spatial/temporal IDs, which are approximately linear signals written into object-word activations at intermediate layers, and validating them with targeted interventions like mirror swapping and steering. It could also shed light on 3D directions and temporal reasoning in videos.

**Strengths:**

1. The problem the paper focus on, spatial reasoning of VLM, is clearly defined and important.
2. The experiments especially in section 2 are clear and well-designed.
3. The discovered spatial IDs show that models confuse "up/down" with "front/back", which helps explain real errors in 3D cases and gives a useful diagnostic. The temporal ID result is similarly clear and interesting.
4. The discovered spatial IDs exist among different VLMs though different backbones and training procedure.

**Weaknesses:**

1. The paper explains how models work in spatial reasoning but does not show that using these IDs can improve benchmark accuracy in a training-free manner or help train better models.
2. The authors could test whether stronger or clearer IDs mean better spatial reasoning ability. For example, by checking the correlation between an "ID strength score" or something similar, and model accuracy on a spatial reasoning benchmark. This would make the finding more practical and convincing.

**Questions:**

Please refer to the weaknesses. I'm glad to raise my score if you cover them well.

---

> ### Author Response · Authors · 2025-11-20
>
> Thank you for your thoughtful review and suggestions! We are glad that you found our problem statement to be “clearly defined and important”, and our experiments to be "clear and well designed". We address your suggestions below:
>
>     “The paper explains how models work in spatial reasoning but does not show that using these IDs can improve benchmark accuracy in a training-free manner or help train better models.”
>
>
> Incorporating spatial IDs to improve benchmark performance of VLMs will certainly strengthen our paper, and highlight the importance of the spatial ID mechanism. We demonstrate how spatial IDs can help model performance through a simple finetuning experiment.
>
> More specifically, we finetune Qwen2-2B on a synthetic dataset similar to the one used to extract spatial IDs, and evaluate on COCO-Spatial. For the spatially informed training, we introduce an additional loss module at layer 11 that computes the cosine similarity between the predicted and ground-truth spatial ID at that layer. This spatial ID loss is added to the standard language modeling objective, providing extra supervision. We also perform a control training where we only use the stanard objective. We revised the manuscript to provide more precise experimental details for this finetuning in **Sec. 4.3** and **Sec. A9**. In summary, we find that indeed, incorporating spatial ID alignment into the finetuning loss allows the model to generalize to the naturalistic validation set quicker.
>
>
> Below is the finetuning result from the control case using just the simple LM loss:
>
> |     Num Steps              | 0    | 800  | 1600  | 2400  | 3200  |
> |-------------------|------|------|-------|-------|-------|
> | LM Loss           | 3.3  | 0.05 | <0.01 | <0.01 | <0.01 |
> | COCO Val Accuracy | 0.77 | 0.83 | 0.84  | 0.85  | **0.85**  |
>
> And below is the result for the combined LM and spatial ID loss:
>
> |          Num Steps         | 0    | 800  | 1600  | 2400  | 3200  |
> |-------------------|------|------|-------|-------|-------|
> | LM Loss           | 2.71 | 0.04 | <0.01 | <0.01 | <0.01 |
> | Spatial ID Loss   | 0.75 | 0.58 | 0.41  | 0.36  | 0.33  |
> | COCO Val Accuracy | 0.77 | 0.83 | 0.84  | 0.88  | **0.91**  |
>
> With the spatial ID loss, the model attains above 90\% accuracy on COCO-spatial at 3.2k steps, which is a 6% absolute advantage over the control case.
> We have added training plots to **Fig. A10** in the supplements. Thank you for this valuable suggestion!
>
>
>
>
>     “The authors could test whether stronger or clearer IDs mean better spatial reasoning ability.”
>
>
> Thank you for the pointer. We verified this with a quick experiment and indeed it seems that stronger spatial IDs correlate with better spatial reasoning capacity! We have added **Fig. 9** in the updated submission, where we see a clear upward trend of evaluation accuracy on COCO-spatial with increasing Spatial-ID "Steerability".
> Below is a summary table of the new figure:
>
> | Model | Steerability (mean ± std) | Accuracy on COCO-spatial |
> |-------|----------------|---------|
> | gemma-4b | 0.15 ± 0.045 | 0.72 |
> | llava-13b | 0.062 ± 0.1 | 0.78 |
> | gemma-12b | 0.2 ± 0.063 | 0.82 |
> | llava-7b | 0.16 ± 0.12 | 0.84 |
> | qwen2-2b | 0.37 ± 0.076 | 0.87 |
> | llama-11b | 0.35 ± 0.13 | 0.88 |
> | internvl-1b | 0.39 ± 0.14 | 0.91 |
> | qwen-3b | 0.58 ± 0.11 | 0.92 |
> | internvl-2b | 0.53 ± 0.11 | 0.94 |
> | internvl-8b | 0.43 ± 0.13 | 0.94 |
> | internvl-14b | 0.56 ± 0.12 | 0.95 |
> | qwen-7b | 0.48 ± 0.14 | 0.96 |
>
> The table has been sorted by increasing accuracy on COCO-spatial. We see that on the whole, models that have stronger spatial IDs, and are therefore steered more readily, have higher accuracy. The linear regression on Steerability vs. Spatial Evaluation Accuracy on these models has a slope of 0.3826, with an R^2 of 0.8018.
>
> For this analysis, we compute the “steerability” as the difference between the change of belief resultant from steering with opposing spatial IDs, minus the change of belief from steering with noise. The layers of intervention are chosen as the middle third of all layers for that model. In **Fig. 9**, each point expresses the mean steerability, and the confidence interval is one standard deviation away from the average. The y axis shows the model’s accuracy on COCO-spatial with no spatial intervention on any layer.
> From this additional analysis, we see that models which learn better organized spatial IDs are more likely to have stronger spatial reasoning performance. This result is in line with how spatial IDs can be a helpful learning signal during finetuning. If better spatial IDs lead to better models, then it makes sense that greater spatial ID alignment will help model performance.
>
> Thank you again for the insightful suggestions!

---

> > ### Author Response · Authors · 2025-11-27
> > **Gentle Reminder of Rebuttals**
> >
> > Thank you again for taking the time to review our paper. We would like to gently remind you that there is just a week left before the rebuttal period deadline. We have added new analyses showing *stronger spatial IDs correlate to stronger models*, and that *using spatial-ID guided loss module promotes faster generalization during fine-tuning*. We believe these new additions demonstrate the usefulness of spatial IDs, and the key role they play in VLM performance.
> >
> > We would greatly value any further feedback or confirmation of whether we have addressed your comments sufficiently.
> > Thank you, and happy almost holidays!

---

### Author Response · Authors · 2025-12-03
**Summary for Area Chair, Reviewers, and Community**

### Paper & Reviews Overview
Our work identifies a linear mechanism through which contemporary VLMs perform spatial and temporal reasoning. We term this mechanism spatiotemporal IDs: internal vectors that bind to an object's *word activation* and encode its location. Using targeted interventions and activation-swapping experiments across multiple SoTA VLMs, we show that these IDs robustly mediate beliefs about object position. We also provide theoretical intuition for why this mechanism arises, demonstrate its relevance for explaining 3D reasoning failures, and diagnose architectural bottlenecks. Replicated analyses in video models reveal an analogous temporal ID mechanism.

Reviewers described the contribution as a “clearly defined and important” problem and an “interesting mechanistic discovery.” They further characterized our experiments as “robust,” “well-designed,” and “rigorous,” with consistent results across diverse SoTA VLMs. Reviewers also highlighted the utility of our mechanism for interpreting 3D errors and localizing model bottlenecks.

We also received great constructive suggestions. We have added several new analyses in response to each comment, and summarize them below:


### Reviews & Added Experiments

- **Sec. 4.3 & Fig. A10 (pp. 9, 20)**: Per suggestions from rYNZ, zD2N, and dAtd, we added a new evaluation showing that spatial-ID–guided loss accelerates generalization during fine-tuning.
- **Fig. 9 (p. 9)**: In response to rYNZ, we show that models more steerable by spatial IDs also score higher on spatial VQA benchmarks.
- **Fig. 6 (p. 6, revised)**: Per zD2N’s request, we expanded our taxonomy of spatial relations to include “near,” “far,” “between,” and related categories.
- **Figs. A6–A7 (p. 18)**: Per suggestions from 4Gni, we show that object-specific attribute reasoning is also mediated by spatial IDs.
- **Figs. A14–A15 (pp. 24–25)**: Per suggestions from dAtd, we added controls replacing non-object tokens in mirror-swapping experiments, confirming that spatial IDs reside in object-token activations.
- **Fig. A9 (p. 19)**: To further support our diagnostic claims as requested by dAtd, we introduced an *oracle injection* setting, which verifies that LLaVA’s bottleneck lies in its image encoder.

These additions, explained in our rebuttal threads and elaborated in our revised manuscript, highlight the exciting breadth of applications enabled by spatiotemporal IDs beyond being a novel mechanistic discovery.

### Final Comments
Although the shortened rebuttal period limited further dialogue, we believe we have thoroughly addressed all reviewer comments. We hope our detailed responses and the reviewers’ stated willingness to adjust scores will be reflected in the final evaluation.

We are grateful to the reviewers and ACs for their thoughtful engagement and for helping to strengthen our work. Happy winter holidays!

---

### Meta-Review · Area_Chair_U9U4 · 2026-01-05

**Summary:**

The key reviewer concerns are:

* Practical utility: show that spatial / temporal IDs are not just descriptive but can help to improve models [rYNZ,zD2N,dAtd]

* Coverage and generality: move beyond simple left/right/up/down to richer relations; extend temporal analysis [zD2N,4Gni]

* Causal attribution and controls: verify IDs are bound to object tokens (e.g. via non-object token controls); disentangle vision encoding vs. cross-modal binding vs. language readout [dAtd]

* Depth vs. vertical conflations: diagnose likely causes and implications [zD2N]

* Clarity of extraction and robustness: clarify how IDs are computed and queries regarding handling object size / extent [4Gni]

As outlined below, the authors provided a constructive rebuttal with new experiments that attempt to directly address the raised queries and concerns. Two reviewers initially lean positive and two lean negative. Reviewer rYNZ explicitly notes that the are open to raising their score, if the rebuttal addresses the raised concerns sufficiently well. In my assessment, the rebuttal mitigates a meaningful portion of the initially negative reviewers’ concerns, sufficient to shift the balance toward acceptance, despite remaining limitations. I recommend accept.

**Reviewer Concerns:**

- Practical utility: authors fine-tune Qwen2-2B with an auxiliary alignment loss and report both somewhat improved generalisation on COCO-Spatial (c.~6% vs. control) in addition to correlation between 'ID steerability' and benchmark accuracy, across multiple VLMs. I consider these updates can go some way towards supporting diagnostic and predictive utility claims.

- Coverage and generality: additional interventions extend relations to 'near/far' and 'in-between', towards showing consistent steering effects. Authors further report that object-attribute binding (colour) can be mediated by spatial IDs. This provides initial evidence that can link broader visual reasoning tasks to the mechanism. Authors concede that Temporal ID considerations are of lesser importance in the current work.

- Causal attribution: authors are keen to reinforce that IDs are bound to object-token activations. Fine-grained controls that replace non-object tokens can show an order-of-magnitude smaller belief change than swapping object tokens. The oracle injection experiment indicating disparities between LLaVA, LLaMA looks to explain component-level fault attribution is less convincing and might benefit from deeper exploration.

- Additional smaller detail clarifications: authors detail their batch-based extraction and show centroid-based locations somewhat capture IDs across object sizes and claim that size is likely encoded along separate directions.

Remaining limitations

- Temporal IDs, while supported, remain less fully explored than spatial IDs; richer temporal benchmarks and interventions would strengthen claims.

- The auxiliary-loss fine-tuning result is demonstrated on a single backbone/setting; broader replication (models/tasks) would solidify generality.

- Depth / vertical diagnosis points to likely data bias; direct mitigation (e.g., targeted data or 3D-aware training) is proposed but not evaluated here.

**Reviewer Scores:**

- Reviewer rYNZ (4): Asked for utility beyond explanation and correlation with performance. The added fine-tuning signal and steerability–accuracy correlation directly address these, making an upward adjustment plausible.

- Reviewer zD2N (6): Sought broader relation coverage and actionable improvements. Authors added relations, utility via auxiliary loss, and discussion of depth conflation. Score likely stable at 6.

- Reviewer 4Gni (6): Requested deeper temporal analysis and whether non-spatial attributes bind via IDs. I can consider new temporal orthogonality and colour-binding results to partially satisfy. Score likely stable at 6.

- Reviewer dAtd (4): Asked for non-object-token controls and clearer causal localisation. New controls and oracle injection can somewhat strengthen causal claims. The reviewer also flagged remediation breadth and may have been less likely to upgrade.

Decision recommendation

This work offers a reasonable contribution that is validated across multiple VLMs with rigorous interventions. I note that the rebuttal adds evidence of practical utility (auxiliary-loss fine-tuning), some nascent cross-model predictive value (steerability correlation), expanded relation coverage, and somewhat improved causal attribution. While further temporal expansion and broader fine-tuning replications would add weight, the contribution appears timely for interpretability and potentially principled VLM improvement. I recommend accept.

---

### Decision · Program_Chairs · 2026-01-26

Accept (Poster)